# Prevalence of Negative Transfer in Continual Reinforcement Learning: Analyses and a Simple Baseline

**Hongjoon Ahn**[1]*, **Jinu Hyeon**[2]*, **Youngmin Oh**[3], **Bosun Hwang**[3], **and Taesup Moon**[4]†

[1] Department of Electrical and Computer Engineering (ECE), Seoul National University,
[2] Interdisciplinary Program in Artificial Intelligence (IPAI), Seoul National University,
[3] Samsung Advanced Institute of Technology,
[4] Department of ECE / IPAI / ASRI / INMC, Seoul National University
{hong0805, kanasimy}@snu.ac.kr,
{youngmin0.oh, bosun.hwang}@samsung.com, tsmoon@snu.ac.kr

## Abstract

We argue that the *negative transfer* problem occurring when the new task to learn arrives is an important problem that needs not be overlooked when developing effective Continual Reinforcement Learning (CRL) algorithms. Through comprehensive experimental validation, we demonstrate that such issue frequently exists in CRL and cannot be effectively addressed by several recent work on either mitigating *plasticity loss* of RL agents or enhancing the positive transfer in CRL scenario. To that end, we develop **R**eset & **D**istill (**R&D**), a simple yet highly effective baseline method, to overcome the negative transfer problem in CRL. R&D combines a strategy of resetting the agent's online actor and critic networks to learn a new task and an offline learning step for distilling the knowledge from the online actor and previous expert's action probabilities. We carried out extensive experiments on long sequence of Meta World tasks and show that our simple baseline method consistently outperforms recent approaches, achieving significantly higher success rates across a range of tasks. Our findings highlight the importance of considering negative transfer in CRL and emphasize the need for robust strategies like R&D to mitigate its detrimental effects. The code implementation is available at https://github.com/hongjoon0805/Reset-Distill.git

## 1 Introduction

Following the impressive recent success of reinforcement learning (RL) (Mnih et al., 2013; Silver et al., 2016; Mnih et al., 2015; OpenAI et al., 2020) in various applications, a plethora of research has been done in improving the learning efficiency of RL algorithms. One important avenue of the extension is the Continual Reinforcement Learning (CRL), in which an agent aims to continuously learn and improve its decision-making policy over sequentially arriving tasks without forgetting previously learned tasks. The motivation for such extension is clear since it is not practical to either re-train an agent to learn multiple tasks seen so far or train a dedicated agent for each task whenever a new task to learn arrives. The need for CRL is particularly pressing when the sequentially arriving tasks to learn are similar to each other as in robot action learning (Kober et al., 2013).

In general, one of the main challenges of continual learning (CL) is to effectively transfer the learned knowledge to a new task (*i.e.*, improve plasticity) while avoiding catastrophic forgetting of previously learned knowledge (*i.e.*, improve stability). So far, most of the CRL methods (Mendez et al., 2020; 2022; Rolnick et al., 2019; Wolczyk et al., 2022) also focus on addressing such a challenge, largely inspired by the methods developed in the supervised learning counterparts; *e.g.,* improving the stability by regularizing the deviation of the important parameters (Kirkpatrick et al., 2017; Zenke et al., 2017; Ahn et al., 2019; Jung et al., 2020), storing the subset of dataset on previous tasks (Chaudhry et al., 2019a;b; Lopez-Paz & Ranzato, 2017) or isolating the important parameters (Mallya & Lazebnik, 2018; Mallya et al., 2018; Hung et al., 2019; Yoon et al., 2018). Furthermore, several works mainly focused on improving the plasticity of the network by transferring the knowledge from

---

*Equal contribution.
†Corresponding author

previous tasks (Rusu et al., 2016; Schwarz et al., 2018) or selectively re-using the important parts for learning new tasks (Mendez et al., 2022; 2020; Mendez & Eaton, 2021).

Due to the aforementioned trade-off, it is generally understood that the plasticity degradation occurs in continual learning mainly due to the emphasis on stability. However, several recent work pointed out that, particularly in RL, the plasticity of a learner can decrease even when learning a *single* task (Nikishin et al., 2022; Lewandowski et al., 2023; Kumar et al., 2021; Lyle et al., 2022; 2023; Sokar et al., 2023; Berariu et al., 2021), in which the stability is not considered at all. Those works identified that the occurrence of such *plasticity loss* may be largely due to using non-stationary targets while learning the value function. These findings give some clues for understanding the plasticity degradation phenomenon in CRL, which occurs quite often not only when learning each task but also when task transition happens, but not the full explanation.

Namely, in CRL, even when the simple *fine-tuning* is employed for sequentially learning tasks, it is not hard to observe that a learner already suffers from learning a new task as we show in our experiments in later sections. We may attempt to explain this plasticity degradation of fine-tuning, which does not consider stability whatsoever, through the lens of the plasticity loss mentioned above; *i.e.,* since the non-stationarity of the learning objectives (or the reward functions) arises when task transition happens, the plasticity loss occurs and hampers the learning ability. However, as we observe from our careful empirical analyses, above explanation is not fully satisfactory since such plasticity degradation turns out to be *dependent* on what specific task a learner has learned previously. That is, we show that the dissimilarity between the learned tasks also becomes a critical factor for the plasticity degradation (of fine-tuning) in CRL, which we identify as the *negative transfer* problem that has been also considered in conventional transfer learning literature (Zhang et al., 2022; Taylor & Stone, 2009; Chen et al., 2019).

To that end, we mainly focus on and try to address the negative transfer problem in CRL. In Section 3, we first carry out a simple three-task experiment that exhibits a severe negative transfer for fine-tuning. We show that simple adoption of the various remedy for the plasticity loss in RL agents proposed in recent works (Nikishin et al., 2022; Lewandowski et al., 2023; Kumar et al., 2021; Lyle et al., 2022; 2023; Sokar et al., 2023; Berariu et al., 2021) cannot successfully mitigate the negative transfer in our setting. Moreover, we also demonstrate that when such negative transfer phenomenon prevails, methods that promote *positive transfer* (beyond fine-tuning) can also result in detrimental results. Subsequently, via more extensive experiments using the Meta World (Yu et al., 2020), DeepMind Control Suite (Tassa et al., 2018), and Atari-100k (Mnih et al., 2013; Kaiser et al., 2020) environments, we identify that various levels of negative transfer exist depending on the task sequences and RL algorithms. From these findings, in Section 4, we propose a simple method, dubbed as **R**eset & **D**istill (**R&D**), that is tailored for CRL and prevents both the negative transfer (via resetting the online learner) and forgetting (via distillation from offline learner). Finally, in Section 5, we present experimental results on longer task sequences and show R&D significantly outperforms recent CRL baselines as well as methods that simply plug-in the recent plasticity loss mitigation schemes to the CRL baselines. The quantitative metric comparisons show the gain of R&D indeed comes from addressing both negative transfer and forgetting. We stress that our result underscores addressing negative transfer phenomenon is indispensable in CRL since our simple baseline method can already surpass the methods that try to promote positive transfers between tasks.

## 2 BACKGROUND

### 2.1 PRELIMINARIES

**Notations.** In CRL, an agent needs to sequentially learn multiple tasks without forgetting the past tasks. We denote the task sequence by a task descriptor $\tau \in \{1, ..., T\}$, in which $T$ is the total number of tasks. At each task $\tau$, the agent interacts with the environment according to a Markov Decision Process (MDP) $(\mathcal{S}_\tau, \mathcal{A}_\tau, p_\tau, r_\tau)$, where $\mathcal{S}_\tau$ and $\mathcal{A}_\tau$ are the set of all possible states and actions for task $\tau$. Given $s_{t+1}, s_t \in \mathcal{S}_\tau$ and $a_t \in \mathcal{A}_\tau$, $p_\tau(s_{t+1}|s_t, a_t)$ is the probability of transitioning to $s_{t+1}$ given a state $s_t$ and action $a_t$. $r_\tau(s_t, a_t)$ is the reward function that produces a scalar value for each transition $(s_t, a_t)$. The objective of an RL agent is to obtain a policy $\pi(a_t|s_t)$ that can maximize the sum of expected cumulative rewards for each task $\tau$.

**RL setting.** In this paper, we mainly focus on the actor-critic method which combines both value-based and policy-based methods. This method includes two networks: an *actor* that learns a policy

and a *critic* that learns the value function; the critic evaluates the policy by estimating the value of each state-action pair, while the actor improves the policy by maximizing the expected reward. Given task $\tau$ and $s_t \in \mathcal{S}_\tau, a_t \in \mathcal{A}_\tau$, we denote the actor parameterized by $\boldsymbol{\theta}_\tau$ as $\pi(a_t|s_t; \boldsymbol{\theta}_\tau)$, and the critic parameterized by $\boldsymbol{\phi}_\tau$ as $Q(a_t, s_t; \boldsymbol{\phi}_\tau)$. For the algorithms that only use the state information in the critic, we denote the critic network as $V(s_t; \boldsymbol{\phi}_\tau)$. In our study, we adopt SAC (Haarnoja et al., 2018) and PPO (Schulman et al., 2017) as representative actor-critic methods due to their stability and efficiency.

## 2.2 Loss of plasticity in RL

Here, we outline recent studies that pointed out the *plasticity loss* of RL algorithms from several different viewpoints. Igl et al. (2021) found an evidence that using the non-stationary target when learning the value function, unlike the stationary target of supervised learning, can permanently impact the latent representations and adversely affect the generalization performance. From a similar perspective, Kumar et al. (2021) and Lyle et al. (2022) figured out that the non-stationarity of the target may diminish the rank of the feature embedding matrix obtained by the value network. They hypothesize that this phenomenon ultimately results in the capacity loss of the value function and hinders the function from learning new tasks. To address this issue, Lyle et al. (2022) proposed a regularization method, *InFeR*, to preserve the rank of the feature embedding matrix. Nikishin et al. (2022) considered another viewpoint and demonstrated that RL methods that tend to highly overfit to the initial data in the replay buffer can suffer from primacy bias that leads to the plasticity degradation for the incoming samples. Furthermore, Sokar et al. (2023) argued that the large number of dormant neurons in the value network, which could be caused by using the non-stationary targets for learning, maybe another reason for the plasticity loss. To address this issue, they proposed *ReDo* that selectively resets the dormant neurons to enlarge the capacity of the network.

While above proposals certainly made some progress, they still remained to be partial explanation for the plasticity loss. Namely, Lyle et al. (2023) showed that as opposed to the analyses in Kumar et al. (2021) and Lyle et al. (2022), the high correlation between the rank of the feature embedding matrix and the plasticity loss only appears when the underlying reward function is either easy or hard to learn. For example, they showed that if the environment produces the sparse rewards, there is low correlation between the feature rank and the plasticity loss. Subsequently, they also showed that the large number of dormant neurons affected the plasticity loss only when the underlying network architecture happens to be multi-layer perceptron. Lyle et al. (2023) proposed a new insight that the root cause of the plasticity loss is the loss of curvature in the loss function. Lewandowski et al. (2023) also stressed that the optimization landscape has diminishing curvature and proposed *Wasserstein regularization* that regularizes the distribution of parameters if it is far from the distribution of the randomly initialized parameters. Lee et al. (2023) divided the plasticity into two aspects. One is the input plasticity which implies the adaptability of the model to the input data, and the other is the label plasticity which implies the ability of the model to adapt to evolving input-output relationship. Lee et al. (2023) show that combining all the methods (e.g. layer normalization, sharpness aware minimization (SAM) (Foret et al., 2021), and reset (Nikishin et al., 2022) that improve the input and label plasticity can enhance the overall plasticity. To the end, Nauman et al. (2024) broadly analyzed various regularization techniques for improving the plasticity, and figure out that resetting the network surpasses other schemes. In Nikishin et al. (2022), Lee et al. (2023) and Nauman et al. (2024), all of them show the effectiveness of the resetting the network while learning a single task. However, since resetting the network can cause complete forgetting of past tasks in CRL setting, naively applying the resetting schemes in CRL would be counterintuitive.

Alternatively, Dohare et al. (2021) considered the degradation of the plasticity of stochastic gradient descent in both continuous supervised and reinforcement learning. Furthermore, Abbas et al. (2023) provided empirical results showing that as a learner repeatedly learns a task sequence multiple times, the performance of each task degrades. They proposed that when the loss of plasticity occurs, the weight change of the value function network consistently shrinks as the gradient descent proceeds. To address this issue, they adopted *Concatenated ReLU (CReLU)* to prevent the gradient collapse. Despite the difference, the authors have referred to this phenomenon as plasticity loss as well.

## 2.3 Negative transfer in transfer learning

The *negative transfer* problem has been identified as one of the important issues to consider in transfer learning (Taylor & Stone, 2009). Namely, in Wang et al. (2019), Cao et al. (2018), Ge et al. (2014) and Rosenstein et al. (2005), they observed that when the source and target domains are not

sufficiently similar, the transfer learned model on the target task may perform even worse than the model that learns the target task from scratch (*i.e.*, negative transfer occurs). In CRL, one may argue that such phenomenon is just another version of plasticity loss mentioned in the previous subsection since the task transition causes the non-stationarity of targets for learning an agent. However, as we show in the next section, our simple experimental results demonstrate that merely applying the methods in Lyle et al. (2022), Abbas et al. (2023) and Lewandowski et al. (2023) that aim to address the plasticity loss issue in RL do *not* readily resolve the negative transfer problem in CRL.

# 3 THE NEGATIVE TRANSFER IN CRL

## 3.1 A MOTIVATING EXPERIMENT

In this section, we carry out a simple experiment on a popular Meta World (Yu et al., 2020) environment, which consists of various robotic manipulation tasks, to showcase the negative transfer problem in CRL.

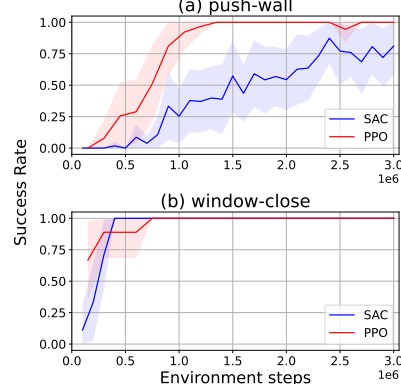

Figure 1: The success rates of SAC and PPO on (a) `push-wall` and (b) `window-close` tasks.

**Existence of negative transfer.** Firstly, Figure 1(a) and (b) show the success rates for learning the `push-wall` and `window-close` tasks with SAC (Haarnoja et al., 2018) and PPO (Schulman et al., 2017) algorithms for 3 million (M) steps from scratch, respectively. Note that both algorithms achieve success rates close to 1, showing both tasks are quite easy to learn from scratch. Now, Figure 2(a) shows the results for continuously learning `sweep-into`, `push-wall` and `window-close` tasks with simple *fine-tuning* (red) for 9M steps (3M steps for each task). Namely, SAC and PPO are simply fine-tuned to `push-wall` after learning `sweep-into` and `window-close` after `push-wall`. In this context, fine-tuning refers to adjusting all parameters of the network without imposing any freezing or regularization constraints, hence, it does not put any emphasis on the stability to combat catastrophic forgetting. In the results, we clearly observe that both SAC and PPO completely fail to learn `push-wall` after learning `sweep-into` even when the fully plastic fine-tuning is employed. Hence, we note such failure cannot be attributed to the well-known stability-plasticity dilemma in continual learning.

**Mitigating plasticity loss cannot fully address negative transfer.** For an alternative explanation, we can check whether such a failure can be identified by the indicators of the plasticity loss developed by the studies presented in Section 2.2. Figure 2(b) shows the number of dormant neurons (Sokar et al., 2023), rank of the feature embeddings (Lyle et al., 2022; Kumar et al., 2021), weight deviation (Abbas et al., 2023), and Hessian sRank (Lewandowski et al., 2023) of the fine-tuned model's actor and critic across the three tasks. When focusing on the `push-wall` task, we observe mixed results; namely, while some indicators (*i.e.,* high dormant neurons and low feature rank) indeed point to the plasticity loss of the model, the others (*i.e.,* high Hessian sRank and high weight deviations) are contradicting. Furthermore, in Figure 2(a), we also plot the results of the methods – *i.e.,* ReDO, InFeR, CReLU, and Wasserstein Regularization – that aim to mitigate the plasticity loss of the model from the perspective of each respective indicator with the same color code in Figure 2(b). Still, the success rates of all methods on `push-wall` remain significantly lower than the one in Figure 1.

Based on these results, we note that the dramatic performance degradation of the fine-tuning model on the `push-wall` task cannot be well explained by the previous work on identifying and mitigating the plasticity loss of an RL agent. Furthermore, if the plasticity loss were truly occurring at the task transitions, the high success rate of the third, `window-close` task cannot be well understood, either[1]. Therefore, we argue that the learnability of an RL task may depend on the preceding task, and the *negative transfer* from the preceding task, which cannot be solely captured by previous research, is one of the main obstacles to overcome in CRL.

**Promoting positive transfer cannot address negative transfer.** There are several works that aim to promote positive transfers between tasks in CRL (beyond simple fine-tuning) (Schwarz et al., 2018; Mendez et al., 2020; Mendez & Eaton, 2021; Mendez et al., 2022). One may expect those

---

[1]While the results vary depending on the learning algorithms, it is still clear that the third task has much higher success rates than the second task.

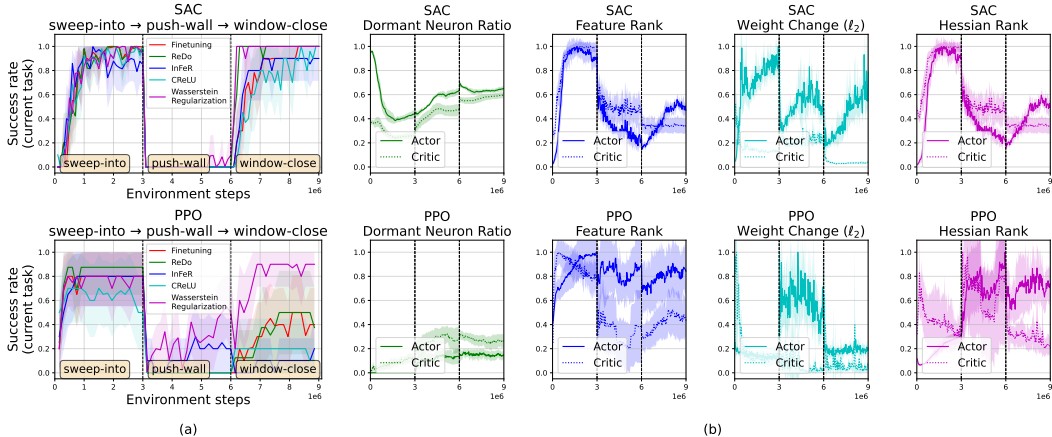

Figure 2: Results on continual fine-tuning SAC (top) and PPO (bottom) on 3 tasks. (a) Success rates with various methods. (b) Various indicators of the plasticity loss of the models across the three tasks.

methods to resolve the negative transfer issue by transferring useful knowledge from the previous task while learning a new task. However, from a simple ablation study on Progress & Compress (P&C) (Schwarz et al., 2018), a well-known baseline of CRL that promotes positive transfer, we observe that such methods also show detrimental performance when there is a negative transfer phenomenon.

More specifically, Figure 3 shows the results of SAC on the same three tasks as in Figure 2 when combined with several variations of P&C. Namely, P&C employs an **adaptor** to promote positive transfer from previous task, and we evaluated the schemes with (w/) and without (w/o) the adaptor. Moreover, as the original P&C paper (Schwarz et al., 2018) has also pointed out, the knowledge learned in the active column and the adaptor may hinder the learning of new incoming task; thus, in addition to the original 'without (w/o) reset' mode, we also tested with (w/) reset, which randomly initializes the network parameters of both active column and adapter when learning a new task begins. Hence, the 'w/ adaptor & w/o reset' and 'w/ adaptor & w/reset' modes are the variations of P&C originally proposed in Schwarz et al. (2018).

From the figure, we first observe that the 'w/o reset' mode, regardless of whether to use the adaptor or not, still fails to learn the second task, `push-wall`. This result shows that the mechanism for promoting positive transfer is not helpful at all, if not harmful, for resolving the severe negative transfer. Secondly, the 'w/reset & w/adaptor' mode slightly increases the performance on `push-wall`, but it still achieves much lower performance than learning from scratch. This suggests that the mechanism for promoting positive transfer may in fact hurt the CRL performance when the degree of the negative transfer is severe. In such a case, addressing the negative transfer may have a higher priority than promoting positive transfer.

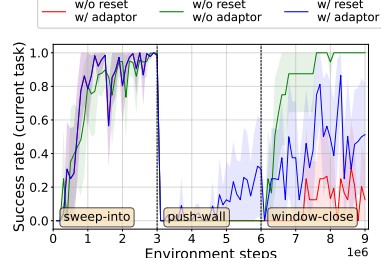

Figure 3: Results of the 3-task experiment with P&C variants, utilizing SAC.

In summary, we show that methods for both mitigating plasticiy loss and promoting positive transfer cannot successfully address the severe negative transfer in our three-task example.

### 3.2 IDENTIFYING VARIOUS LEVELS OF NEGATIVE TRANSFERS

Motivated by the previous subsection, we carried out more extensive experiments using the Meta World (Yu et al., 2020), DeepMind Control Suite (Tassa et al., 2018), and the Atari-100k (Mnih et al., 2013; Kaiser et al., 2020) environment to check the various patterns of negative transfer in CRL.

**Meta World.** We carefully selected 24 tasks that can be successfully learned from scratch, *i.e.,* that can achieve success rates close to 1, within 3M steps. We then categorized them into 8 groups by grouping the tasks that share the same first word in their task names. The 8 task groups were {`Button`, `Door`, `Faucet`, `Handle`, `Plate`, `Push`, `Sweep`, `Window`}, and for more details on the specific tasks in each group, please refer to the Appendix B. Note that the groups were constructed

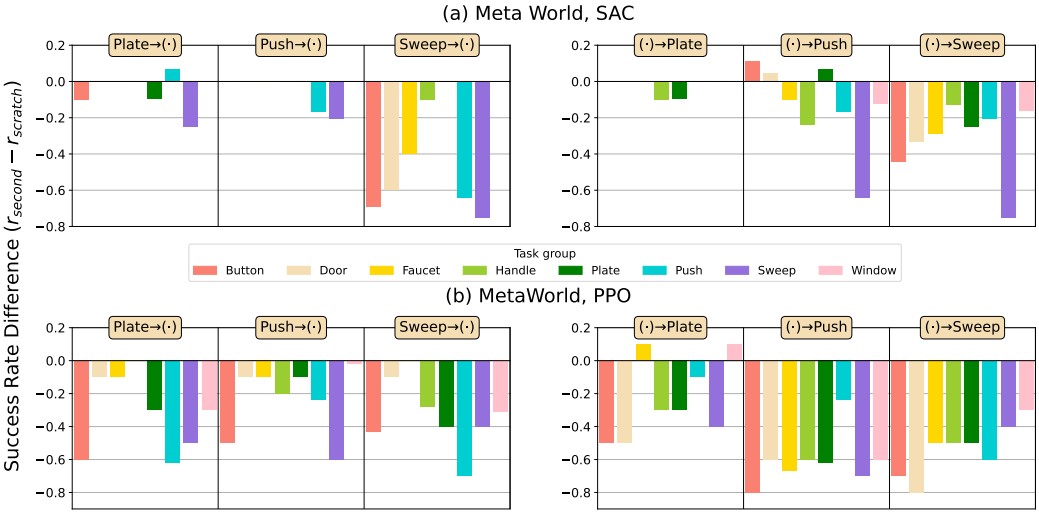

Figure 4: Negative transfer patterns for the two-task fine-tuning in Meta World with (a) SAC and (b) PPO, when tasks from `Plate`, `Push` and `Sweep` groups are learned as the first (left) or the second (right) task.

simply based on the names of the tasks, hence, the tasks that are in the same group can also be largely dissimilar — the main reason for the grouping is to save computational cost for our experiments.

After the task grouping, we carried out substantial two-task CRL experiments with fine-tuning as shown in Figure 4. Namely, we first picked three groups, `Plate`, `Push`, and `Sweep`, and verified the patterns of the negative transfer on the second tasks depending on (i) when they come as the first task, (ii) when they come as the second task, and (iii) when the applied RL algorithm varies. More specifically, the left two figures in Figure 4 are for the results when tasks from `Plate`, `Push`, or `Sweep` task group come as the first task and show the learnability degradation in the 8 task groups that come as the second task (*i.e.,* case (i)). In order to save the computation for the experiments, we did not carry out the exhaustive pairwise two-task experiments, but averaged the results of the following randomized experiments. Namely, we randomly sampled tasks from the first and second task groups and sequentially learned those tasks with 3M steps each with fine-tuning, for 10 different random seeds. When the first and second task groups are identical, we sampled two *different* tasks from the group and carried out the two-task learning. Then, we computed the average of the differences of $r_{\text{second}}$, the success rate of the second task learned by fine-tuning after learning the first task, and $r_{\text{scratch}}$, the success rate of learning the second task from scratch, for each second task group. The average was done over the number of episodes and random seeds, and the more negative difference implies the severer negative transfer. The right two figures are for the reverse case, *i.e.,* when tasks from `Plate`, `Push`, or `Sweep` task group come as the second task, the average success rate differences are depicted depending on the first task group (*i.e.,* case (ii)) [2]. Finally, the upper and lower figures are for the two popular RL algorithms, SAC and PPO (*i.e.,* case (iii)). Overall, we did 10 (random seeds) × 39 (two-task pairs) × 2 (algorithms) = 780 two-task experiments.

From the figures, we can first observe that PPO tends to suffer from the negative transfer more severely than SAC in general. Furthermore, it is apparent that the negative transfer pattern differs depending on the specific task sequence. Namely, for the `Plate` task group with SAC, the negative transfer rarely occurs regardless of the task group being the first or second tasks. However, for the `Push` group with SAC, we observe that while the tasks in the group do not cause too much negative transfer on the second tasks when they are learned first, they tend to suffer from negative transfer when learned after other tasks. Finally, for the `Sweep` group with SAC, it is evident that the tasks in the group both cause negative transfer on the second tasks and suffer from the negative transfer from the first tasks.

**DM Control.** In this experiment, we selected 7 tasks, which are {`ball-in-cup-catch`, `cartpole-balance`, `cartpole-swingup`, `finger-turn-easy`, `fish-upright`, `point-mass-easy`, `reacher-easy`}. Using those tasks, we trained SAC and PPO, and similar to the

---

[2]The experiments for the overlapping pairs as in case (i) are *not* repeated, but the same results are shown.

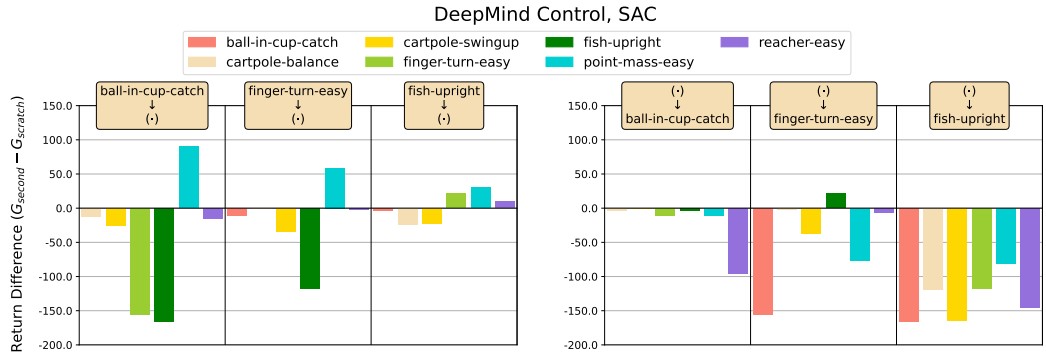

Figure 5: Negative transfer patterns in DeepMind Control environment with SAC.

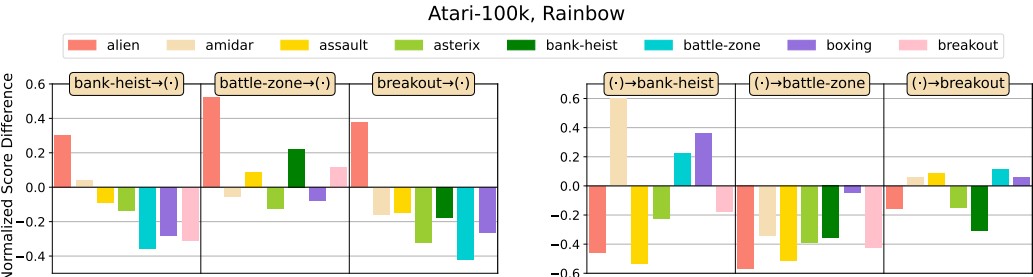

Figure 6: Negative transfer patterns in Atari-100k environment with Rainbow.

case in the Meta World experiment, we also selected 3 representative tasks, {`ball-in-cup-catch`, `finger-turn-easy`, `fish-upright`}. Here, we did not make task groups. Figure 5 shows the results of the SAC. To measure the degree of negative transfer, we report the return difference. In the figure, the negative transfer also occurs quite frequently, and especially for `fish-upright` task, this task always suffer from the negative transfer regardless of the first task. This phenomenon is similar to `push` task group in the Meta World. For more details, please refer to Appendix I

**Atari.** Different from Meta World and DM Control, Atari environment produces visual observation, which is much more complex than previous experiments. In this experiment, we selected 8 tasks, which are {`alien`, `assault`, `bank-heist`, `boxing`, `amidar`, `asterix`, `battle-zone`, `breakout`}. In this experiment, we trained Rainbow (Hessel et al., 2018) on two task pairs, and we selected 3 representative tasks, {`bank-heist`, `battle-zone`, `breakout`}. Figure 6 shows the results. We report the difference of the normalized score which is normalized by the score obtained from scratch. In the figure, we can clearly observe that the negative transfer problem also occurs between Atari games. Especially for the `battle-zone`, when it comes as the second task, it suffer from severe negative transfer in most cases. For more details, please refer to Appendix J

## 4 A SIMPLE BASELINE FOR ADDRESSING THE NEGATIVE TRANSFER IN CRL

**Motivation.** In the previous section, it was highlighted that in cases where negative transfer is severe, the previous knowledge from earlier tasks can become an obstacle to learning. To mitigate the effects of negative transfer, it is essential to consider two key factors; first, all prior knowledge from previous tasks should be erased during the learning of the current task, and second, the catastrophic forgetting must be prevented to enable sequential learning across multiple tasks. A straightforward approach to eliminating previous knowledge is to *randomly re-initialize* all network parameters. However, this is insufficient on its own, as it inevitably leads to forgetting within the network, making it difficult to meet both requirements. To overcome this challenge, we propose the use of two actor networks: the online and offline learners. This dual-network framework with periodic reset aims to balance the need to discard outdated knowledge to learn the current task while maintaining the ability to learn sequentially without forgetting. Our method can be considered analogous to the 'w/reset, w/o adaptor' mode in the experiment shown in Figure 3. However, while P&C applies dual network mechanism in

conjunction with an adaptor to facilitate positive transfer, our primary motivation lies in reducing transfer itself. A more detailed discussion on the comparison with P&C is provided in Appendix F

**Reset and Distill (R&D)** To describe our method in details, we denote the parameters of online actor and critic network as $\boldsymbol{\theta}_{\text{online}}$ and $\boldsymbol{\phi}_{\text{online}}$, respectively, and the parameters of offline actor as $\boldsymbol{\theta}_{\text{offline}}$. We periodically reset $\boldsymbol{\theta}_{\text{online}}$ and $\boldsymbol{\phi}_{\text{online}}$ after finishing learning each task. For the notational convenience, we denote the parameters of the online actor right after learning task $\tau$, but before resetting them, as $\boldsymbol{\theta}_{\tau}^{*}$. Note these parameters are not stored or utilized directly in the subsequent tasks.

Based on the notation, consider when $\tau = 1$, *i.e.*, the learning the first task. Clearly, the online actor and critic can learn the task with existing RL algorithms like SAC or PPO. Once the learning is done, we can then generate a replay buffer $\mathcal{D}_\tau$ by utilizing the expert actor with parameter $\boldsymbol{\theta}_{\tau}^{*}$[3]. Next, we train the offline actor using the state-action pairs in $\mathcal{D}_\tau$ by *distilling* the knowledge from $\boldsymbol{\theta}_{\tau}^{*}$ to the offline actor with $\boldsymbol{\theta}_{\text{offline}}$. Then, we store $\mathcal{M}_\tau$, a small subset of $\mathcal{D}_\tau$, in the expert buffer $\mathcal{M}$. After completing the training for the initial task, we *reset all of the parameters* $\boldsymbol{\theta}_{\text{online}}$ and $\boldsymbol{\phi}_{\text{online}}$ before initiating learning for the next task. The whole process is iteratively applied to subsequent tasks $\tau = 2, \cdots, T$. During the distillation process after the first task, the buffer for the current task, $\mathcal{D}_\tau$, and the buffer containing information from all previously encountered tasks, $\mathcal{M}$, are used together to prevent forgetting. Hence, the loss function for the offline actor for task $\tau$ becomes

$$\ell_{\textbf{R\&D},\tau}(\boldsymbol{\theta}_{\text{offline}}) = \underbrace{\sum_{(s_t,\pi_\tau)\in\mathcal{B}_{\mathcal{D}_\tau}} \text{KL}\Big(\pi(\cdot|s_t;\boldsymbol{\theta}_\tau^*)\big\|\pi(\cdot|s_t;\boldsymbol{\theta}_{\text{offline}})\Big)}_{(a)} + \underbrace{\sum_{(s_t,\pi_k)\in\mathcal{B}_{\mathcal{M}}} \text{KL}\Big(\pi(\cdot|s_t;\boldsymbol{\theta}_{\text{offline}})\big\|\pi(\cdot|s_t;\boldsymbol{\theta}_k^*)\Big)}_{(b)},$$

in which $\mathcal{B}_{\mathcal{D}_\tau}$ and $\mathcal{B}_{\mathcal{M}}$ are mini-batch sampled from $\mathcal{D}_\tau$ and $\mathcal{M}$, respectively, $\pi_\tau \triangleq \pi(\cdot|s_t;\boldsymbol{\theta}_\tau^*)$ and $\pi_k \triangleq \pi(\cdot|s_t;\boldsymbol{\theta}_k^*)$, and $k < \tau$ refers to the tasks preceding the training of the current task $\tau$. Note that the term (a) corresponds to the knowledge distillation from the online actor to the offline actor, and the term (b) corresponds to the behavior cloning to prevent the forgetting. The final outcome of this method is the offline actor, $\boldsymbol{\theta}_{\text{offline}}$, which has sequentially learned all tasks. Note our method has two distinct training phases: the first to reset parameters for the online learner, and the second to distill knowledge to the offline learner. Consequently, we dub our algorithm as **R**eset and **D**istill (**R&D**), and a comprehensive summary of the Algorithm is given in Appendix A.

**Potential concerns about R&D.** First, unlike the online learner, one may argue that the offline actor could still suffer from negative transfer. However, we expect the degree of negative transfer to be lower for the offline learner, for which the learning occurs with fixed target labels, than the online learner, which uses the network's outputs as the varying target labels during RL bootstrapping. Second, due to the periodic reset of the online learner, it could be argued that R&D never takes the positive transfer into account. While it is a valid point, as elaborated in Section 3.1, we have observed that the CRL methods that promote positive transfer would often suffer from significant performance degradation due to severe negative transfer rather than gaining benefits of positive transfer. Hence, the primary design goal of R&D was to focus solely on mitigating negative transfer. While positive transfer remains a critical issue in CRL, and R&D cannot be seen as a permanent solution for CRL, if R&D demonstrates superior performance compared to other algorithms, it would indicate that addressing negative transfer should take priority before considering positive transfer.

## 5 EXPERIMENTAL EVALUATION

### 5.1 TWO-TASK FINE-TUNING EXPERIMENTS WITH VARIOUS METHODS

We applied R&D to SAC and PPO on two consecutive tasks to evaluate its effectiveness in mitigating negative transfer, as detailed in Section 3. We then compared these results with fine-tuning, CReLU (Abbas et al., 2023) and InFeR (Lyle et al., 2022) to assess their relative performance. Figure 7 provides the results. In many cases, we can observe that CReLU and InFeR still suffer from the

---

[3]For an off-policy algorithm like SAC, we may reuse the replay buffer to train the online actor and critic. However, such reuse may lead to a degradation in the performance of the offline policy, due to the discrepancy between state-action pairs and the expert. The experimental results on this matter are in Appendix H.

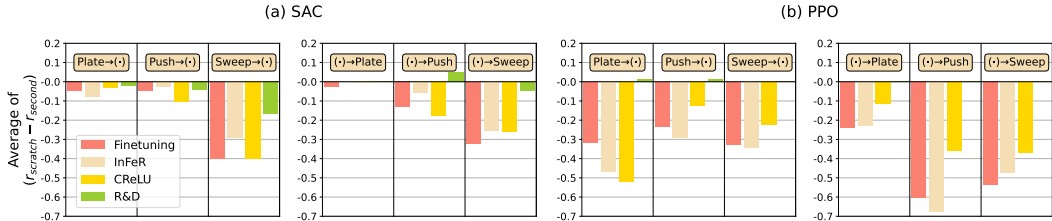

Figure 7: Negative transfer patterns using (a) SAC and (b) PPO with various methods when tasks from `Plate`, `Push` and `Sweep` groups are learned as the first or the second task. For each method, the difference of success rates is averaged over all randomly sampled first or second tasks.

negative transfer in PPO, and R&D effectively resolved the negative transfer. In SAC, the overall performance of R&D for tackling the negative transfer is much better than InFeR and CReLU. The main difference between R&D and the other methods in terms of the learning procedure is the usage of bootstrapping. Different from the RL methods which use bootstrapping for learning the value functions, since R&D learns a new task only through the supervised learning (*i.e.*, the knowledge distillation from the online actor in term (a)), the degree of the negative transfer is much less than the fine-tuning variations.

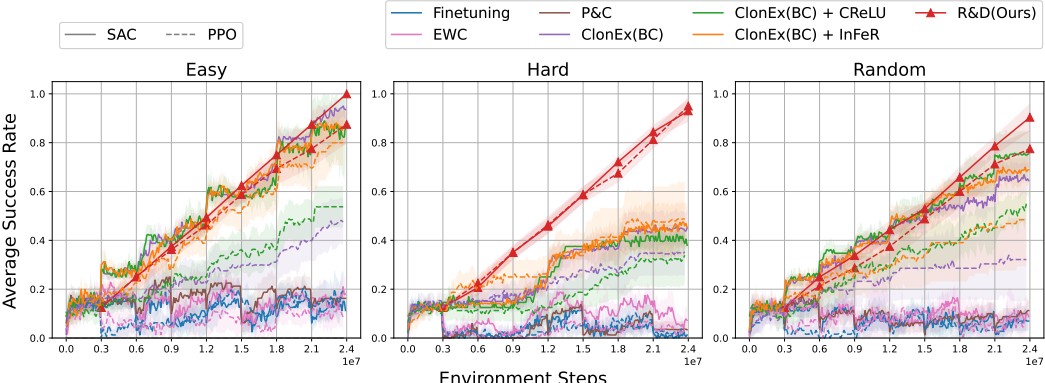

Figure 8: The average success rates of different methods for three types of sequences.

## 5.2 EVALUATION ON LONG SEQUENCE OF TASKS

We also evaluated R&D on long task sequences which consist of multiple environments from Meta World, and compare the results with several state-of-the-art baselines. We used a total of 3 task sequences: Easy, Hard, and Random. For Easy and Hard sequences, the degree of the negative transfer is extremely low and high, respectively. For the Random sequence, we randomly selected and shuffled 8 tasks. For the details on each sequence, please refer to the Appendix C.

The three baselines we used are the following: EWC (Kirkpatrick et al., 2017), P&C (Schwarz et al., 2018), and ClonEx (Wolczyk et al., 2022), along with naïve fine-tuning [4]. Furthermore, we also compare our method to ClonEx with InFeR (Lyle et al., 2022) and CReLU (Abbas et al., 2023) to check whether those methods can tackle both negative transfer and catastrophic forgetting. Note that ClonEx leverages the best-reward exploration technique originally designed only for SAC, leading us to choose Behavioral Cloning (BC) as the method for PPO implementation.

Figure 8 shows the results. Since the offline actor of R&D learns new tasks in an offline way, we instead put markers on the results of R&D and connected them with lines to notice the difference between the baselines. All results are averaged over 10 random seeds. In this figure, we can observe that when the negative transfer rarely occurs ('Easy'), the performances of R&D and ClonEx, ClonEx with InFeR, and ClonEx with CReLU are similar. However, when the negative transfer frequently occurs ('Hard' or 'Random'), R&D outperforms all the baselines.

Table 1: The results on negative transfer and forgetting with various schemes.

| Measure | Negative transfer (↑) | | | Forgetting (↓) | | |
|---|---|---|---|---|---|---|
| Sequence | Easy | Hard | Random | Easy | Hard | Random |
| | SAC / PPO | | | | | |
| Fine-tuning | -0.096 / -0.379 | -0.500 / -0.624 | -0.193 / -0.425 | 0.900 / 0.361 | 0.504 / 0.331 | 0.777 / 0.336 |
| EWC | -0.071 / -0.536 | -0.457 / -0.676 | -0.260 / -0.375 | 0.852 / 0.319 | 0.512 / 0.281 | 0.671 / 0.430 |
| P&C | -0.071 / - | -0.507 / - | -0.207 / - | 0.871 / - | 0.472 / - | 0.702 / - |
| ClonEx | -0.057 / -0.425 | -0.513 / -0.608 | -0.276 / -0.438 | 0.015 / 0.027 | 0.005 / 0.043 | 0.040 / 0.014 |
| ClonEx + CReLU | -0.196 / -0.325 | -0.558 / -0.610 | -0.213 / -0.275 | 0.039 / 0.029 | 0.067 / 0.003 | 0.012 / -0.014 |
| ClonEx + InFeR | -0.117 / -0.075 | -0.503 / -0.462 | -0.232 / -0.286 | 0.031 / 0.043 | 0.001 / -0.014 | 0.038 / 0.000 |
| R&D | **-0.002 / 0.025** | **-0.041 / 0.025** | **-0.014 / 0.013** | 0.000 / 0.050 | 0.008 / 0.029 | 0.045 / 0.029 |

## 5.3 ANALYSES ON NEGATIVE TRANSFER AND FORGETTING

To quantitatively analyze how negative transfer and forgetting actually occurs in our experiments, we measured the forgetting and transfer of 7 methods: R&D, Fine-tuning, EWC, P&C, ClonEx(BC), ClonEx with CReLU, and ClonEx with InFeR. Let us denote the success rate of the task $j$ when the actor immediately finished learning task $i$ as $R_{i,j}$, and the success rate after training task $i$ from scratch as $R_i^{\text{Single}}$. Then the transfer after learning task $\tau$, denoted as $T_\tau$, and the forgetting of task $i$ after learning task $\tau$, denoted as $F_{\tau,i}$, are defined as follows, respectively:

$$T_\tau = R_{\tau,\tau} - R_\tau^{\text{Single}} \quad \text{and} \quad F_{\tau,i} = \max_{l \in \{1,...,\tau-1\}} R_{l,i} - R_{\tau,i}.$$

After learning all $T$ tasks, for the transfer and the forgetting, we report the average of $T_\tau$ and $F_{T,i}$ for all task $\tau \in \{1, ..., T\}$ and $i \in \{1, ..., T\}$, respectively. In this measure, for the transfer, if this has negative value, it indicates the negative transfer occurs. Note that the higher values of transfer and the lower values of forgetting are better in our setting. Table 1 presents the results on the transfer and forgetting, while Appendix L provides the results with standard deviations for reference. In this table, all CRL baselines, except for R&D, display vulnerability to negative transfer. Across all methods, negative transfer tends to be more prominent in the 'Hard' sequence compared to the 'Easy' sequence, whereas it appears to be at a moderate level for the 'Random' sequence. It is worth mentioning that, as discussed in Section 3, PPO exhibits a higher propensity for negative transfer compared to SAC.

In terms of forgetting, it appears that CRL methods, excluding ClonEx and R&D, also experience catastrophic forgetting. Given that SAC typically exhibits greater forgetting than PPO, one might infer that PPO is a more suitable choice for CRL. But this is not the case, as negative transfer rate of PPO is higher than that of SAC, resulting in a smaller number of trainable tasks in the sequence for PPO. Therefore, it is inappropriate to directly compare the forgetting of SAC and PPO.

In our previous findings, we observed that while the average success rate of ClonEx surpasses that of other CRL baselines, it still falls short of the average success rate achieved by R&D. However, the results indicate that ClonEx exhibits forgetting comparable to R&D. Hence, we can deduce that the performance degradation of ClonEx is attributed to negative transfer rather than forgetting.

## 6 CONCLUSION

In this paper, we demonstrate the pervasiveness of negative transfer in the CRL setting. Specifically, we show that recent studies addressing plasticity loss do not effectively mitigate this issue, as evidenced by comprehensive and extensive experiments conducted in the Meta World environment. To effectively address negative transfer in CRL, we propose R&D, a simple yet highly effective method. Experimentally, we illustrate that R&D, utilizing both resetting and distillation, not only addresses negative transfer but also effectively mitigates the catastrophic forgetting problem.

---

[4]Note that because of the severe negative transfer when training P&C with PPO, the loss diverges to infinity, hence, we were unable to train P&C with PPO. Therefore, we only report the results of P&C with SAC.

## ACKNOWLEDGMENTS

This work was supported in part by the National Research Foundation of Korea (NRF) grant [RS-2021-NR059237] and by Institute of Information & communications Technology Planning & Evaluation (IITP) grants [RS-2021-II211343, RS-2021-II212068, RS-2022-II220113, RS-2022-II220959] funded by the Korean government (MSIT). It was also supported by AOARD Grant No. FA2386-23-1-4079 and Samsung Electronics Co., Ltd [Grant No. IO220808-01793-01].

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

## APPENDIX

## A ALGORITHM

---
**Algorithm 1** **R**eset and **D**istill (**R&D**)
---
**Input:** Number of epochs $E$; Total number of tasks $T$
**Initialize:** Network parameters $\boldsymbol{\theta}_{\text{online}}$, $\boldsymbol{\phi}_{\text{online}}$ and $\boldsymbol{\theta}_{\text{offline}}$; Expert buffer $\mathcal{M} \leftarrow \emptyset$
**for** $\tau = 1, \cdots, T$ **do**
   **if** $\tau > 1$ **then**
      *Randomly re-initialize* $\boldsymbol{\theta}_{\text{online}}$ *and* $\boldsymbol{\phi}_{\text{online}}$
   **end if**
   Learn task $\tau$ using $\boldsymbol{\theta}_{\text{online}}$ and $\boldsymbol{\phi}_{\text{online}}$
   Generate replay buffer $\mathcal{D}_\tau$
   **for** $e = 1, \cdots, E$ **do**
      Sample $\mathcal{B}_{\mathcal{D}_\tau} \sim \mathcal{D}_\tau$ and $\mathcal{B}_{\mathcal{M}} \sim \mathcal{M}$
      Compute $\ell_{\textbf{R\&D}}(\boldsymbol{\theta}_{\text{offline}})$ using $\mathcal{B}_{\mathcal{D}_\tau}$ and $\mathcal{B}_{\mathcal{M}}$
      Update $\boldsymbol{\theta}_{\text{offline}}$ with $\nabla\ell_{\textbf{R\&D}}(\boldsymbol{\theta}_{\text{offline}})$
   **end for**
   Store small subset $\mathcal{M}_\tau$ of $\mathcal{D}_\tau$ into $\mathcal{M}$
**end for**
---

## B DETAILS ON 8 TASK GROUPS

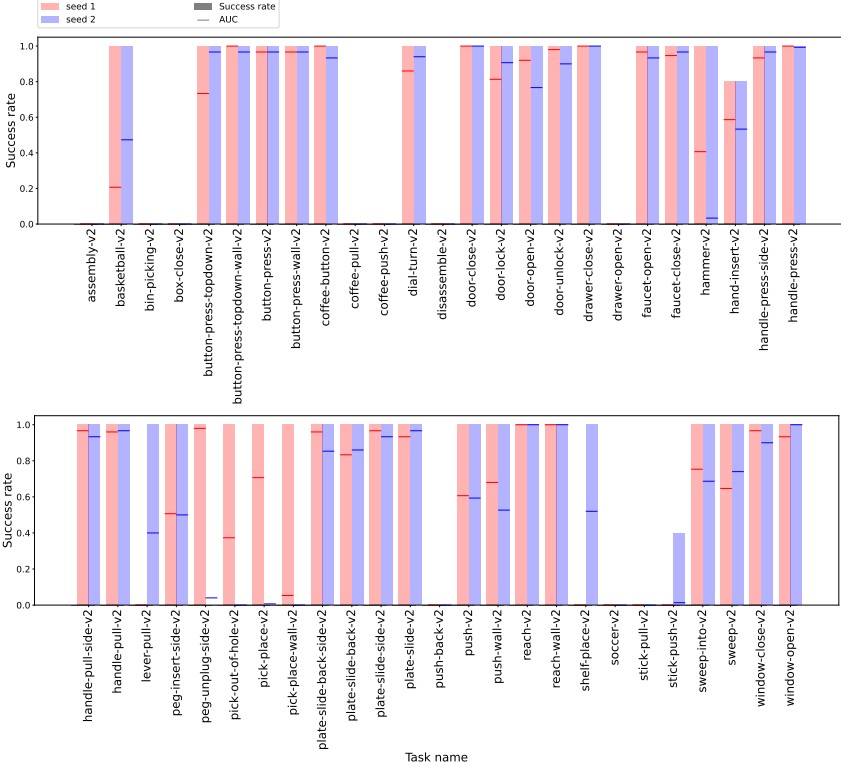

Figure 9: Success rates after training 50 task in Meta-World for 3M steps. SAC was used for training. Results from two different random seeds are distinguished by different colors. The bar plot represents the success rate, and the line marker represents the area under the curve (AUC) of the success rate curve obtained during training.

Prior to examining negative transfer in CRL, we identified tasks that could be learned within 3M steps among the 50 robotic manipulation tasks included in Meta-World Yu et al. (2020).

Figure 9 illustrates the success rates when training the 50 tasks using the SAC algorithm Haarnoja et al. (2018) for 3M steps. In this figure, tasks with lower area under the curve (AUC) values can be interpreted as requiring a relatively larger number of steps for training. This implies that some tasks may not be learned within 3M steps in certain cases. Therefore, to identify negative transfer in specific tasks, it is necessary to prioritize tasks that can be fully learned within 3M steps, i.e., tasks with high success rates and AUC values. Following this criterion, we selected 24 tasks:

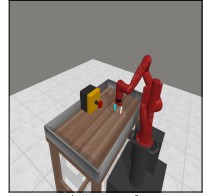 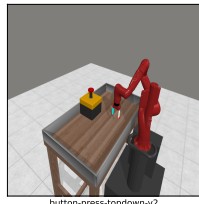

button-press-v2    button-press-topdown-v2

Figure 10: Visualization of button-press (left) and button-press-topdown (right).

As indicated by their names, the tasks can be classified based on similarity. For example, as seen in Figure 10, both `button-press` and `button-press-topdown` involve the robot pressing a button, with the only difference being the direction of the button. By grouping similar tasks together, the 24 selected tasks can be classified into a total of 8 groups.

- **Button**: {button-press-topdown, button-press-topdown-wall, button-press, button-press-wall}
- **Door**: {door-close, door-lock, door-open, door-unlock}
- **Faucet**: {faucet-open, faucet-close}
- **Handle**: {handle-press-side, handle-press, handle-pull-side, handle-pull}
- **Plate**: {plate-slide-back-side, plate-slide-back, plate-slide-side, plate-slide}
- **Push**: {push, push-wall}
- **Sweep**: {sweep-into, sweep}
- **Window**: {window-close, window-open}

## C    DETAILS ON THE LONG SEQUENCE EXPERIMENTS

We evaluated R&D on long task sequences which consist of multiple environments from Meta-World, and compare the results with several state-of-the-art CL baselines. For the experiment, we used a total of 3 task sequences. Firstly, we identified task pairs that exhibit negative transfer when fine-tuning two tasks consecutively. With this information, it is possible to compare the potential difficulties between the task sequences we want to learn. For example, consider different task sequences like A→B→C→D and E→F→G→H where each alphabet represents one task. If we observed negative transfer occurring in consecutive task pairs (A, B), (C, D) and (F, G) within the sequences, the first sequence contains two pairs likely to exhibit negative transfer, while the second has only one such pair. Therefore we can expect the first sequence to be more challenging than the second.

We utilized this method to create two task sequences, each with a length of 8: 'Hard' and 'Easy'. The 'Hard' sequence comprises 6 task pairs where negative transfer occurs in the 2-task setting, while the 'Easy' sequence is generated by connecting only those task pairs where negative transfer does not occur. To further validate the results in an arbitrary sequence, we randomly chose 8 out of the 24 tasks employed in the preceding section and conducted training by shuffling them based on each random seed. Henceforth, we will refer these arbitrary sequences as the 'Random' sequence.

The sequences constructed using as above are as follows. Task name marked in bold indicates that negative transfer may occur when it is learned continuously followed by the previous task.

**Easy** {faucet-open → door-close → button-press-topdown-wall → handle-pull → window-close → plate-slide-back-side → handle-press → door-lock}

**Hard** {faucet-open → **push** → **sweep** → **button-press-topdown** → window-open → **sweep-into** → **button-press-wall** → **push-wall**}

**Random** {door-unlock, faucet-open, handle-press-side, handle-pull-side, plate-slide-back-side, plate-slide-side, shelf-place, window-close}

## D  DETAILS ON THE BASELINES

In this section, we explain the details of the baselines in our experiments.

- EWC (Kirkpatrick et al., 2017): Elastic Weight Consolidation (EWC) is a regularization-based continual learning method in which the Fisher information matrix captures the importance of each weight and gives high regularization strength on important weights.

- P&C (Schwarz et al., 2018): Progress and Compress (P&C) is a regularization-based method that adopts EWC as a policy consolidation scheme and a progressive network to promote forward transfer.

- ClonEx (Wolczyk et al., 2022): ClonEx is a simple yet effective method which performs behavior cloning on previous tasks' policy to increase the stability, and do exploration actively on an arriving task to increase the plasticity. As in our results, ClonEx is effective on preventing the catastrophic forgetting in continual reinforcement learning.

- CReLU (Abbas et al., 2023): Concatenated ReLU (CReLU) is originally designed for object recognition Shang et al. (2016), but the authors of (Abbas et al., 2023) find that the use of CReLU can effectively address plasticity loss in continual reinforcement learning. $CReLU(x) \doteq [ReLU(x), ReLU(-x)]$ is a concatenation of the ReLU outputs of the original input and its negation. Since CReLU can propagate the gradient regardless of the sign of the input, the authors expected that the use of CReLU can promote the weight change which can mitigate the loss of plasticity.

- InFeR (Lyle et al., 2022): (Lyle et al., 2022) show that training the RL agent with the non-stationary target diminishes the numerical feature rank which indicates the target-fitting capacity, and as a result the plasticity loss causes. To tackle this problem, (Lyle et al., 2022) introduce a regularization scheme, InFeR, which regularizes the features of the penultimate layer with the features of randomly initialized network which contains high target-fitting capacity with high feature rank.

Compared to the baselines, the additional memory and computational budget of R&D is marginal. Namely, both EWC (Kirkpatrick et al., 2017) and P&C (Schwarz et al., 2018) also store the networks that learned previous and current tasks, while ClonEx (Wolczyk et al., 2022) stores state samples for computing the behavioral cloning loss. Usually, storing the samples takes much larger memory budget than storing the network parameters. For R&D, it stores both two models and state samples in the buffer. In case of the memory budget, compared to ClonEx, the additional component in R&D is the network for offline actor, and in our experiment, the number of parameters for the networks we used is small. Therefore, R&D does not require a large amount of memory budget compared to the ClonEx. In case of the computational budget, training the online actor for all methods takes 8 hours. For R&D, extracting the rollouts takes 15 minutes and the training time for offline actor takes 25 minutes. Therefore, the additional computational budget for R&D is also small compared to the baselines.

# E   RESULTS OF R&D ON 3-TASK EXPERIMENTS

In this section, we include not only the results of R&D but also the results of the online actor of R&D which corresponds to 'w/reset & w/o adaptor'. Figure 11 shows the results. In the figure, we can observe that the online actor does not suffer from the negative transfer, and eventually, the R&D can also effectively resolve the negative transfer. Through the above results, we want to stress that resetting the whole agent and discarding the previously learned knowledge is effective on tackling the negative transfer.

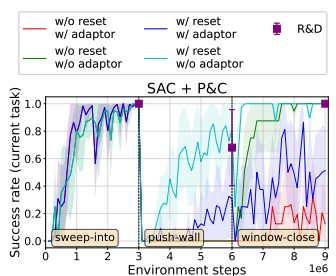

Figure 11: Results of the 3-task experiment with P&C variants and R&D, utilizing SAC

# F   A MORE DETAILED DISCUSSION OF THE DIFFERENCES BETWEEN P&C AND R&D

## F.1   THE ADAPTORS IN P&C

In P&C (Schwarz et al., 2018), there exists a module called an **adaptor**, which serves as an additional mechanism designed to enable the active column to utilize past information stored in the knowledge base. Specifically, the adaptor combines the activations of the active column and the knowledge base in a layer-wise manner, as expressed by the following equation:

$$h_i = \sigma \left( W_i h_{i-1} + \alpha_i \odot U_i \sigma \left( V_i h_{i-1}^{\text{KB}} + c_i \right) + b_i \right)$$

Here, $W_i$ and $b_i$ represent the weight and bias of the active column, while $U_i$, $V_i$, and $c_i$ denote the weight and bias of the adaptor. To compute the activation $h_i$ of the active column, the activation $h_{i-1}^{\text{KB}}$ from the knowledge base is processed through the adaptor. Additionally, the original paper mentions that re-initializing the network parameters of the active column and adaptor allows for the successful learning of a more diverse range of tasks.

## F.2   FUNDAMENTAL DIFFERENCES IN DESIGN AND MOTIVATION BETWEEN P&C AND R&D

While the online/offline actors mechanism in R&D is indeed similar to P&C's implementation of the active column and knowledge base, P&C and R&D differ not only in architecture but also in their fundamentally distinct underlying motivations

A key feature of P&C is the utilization of models trained on previous tasks through lateral connections. To achieve this, adaptor modules, as described above, are extensively employed to facilitate these connections. This is not merely a structural detail; the adaptors serve as essential components for leveraging knowledge acquired from previous tasks to effectively learn the current task, thereby enabling *progress*. In this regard, the knowledge base in P&C is not only a continual learner but also a supporting mechanism that enhances the efficient learning of the active column.

In contrast, R&D does not utilize adaptors or other lateral connections. This indicates that there is no transfer from the offline actor to the online actor, as such transfer can sometimes negatively impact the learning of the current task. We have demonstrated this experimentally in Figure 11. As a result, unlike P&C's knowledge base, the offline actor in R&D functions solely as a continual learner, receiving knowledge distillation from the online actor.

# G  Effect of the size of $\mathcal{M}_\tau$ and $\mathcal{D}_\tau$

In this section, we additionally investigate the impact of the size of the expert and the replay buffer on the performance of R&D. To examine the effect, we conducted experiments by varying the size of the replay buffer used in the distillation phase from 10k to 1M, and the size of the expert buffer from 1k to 10k. We used 'Hard' sequence, which can be considered as the most challenging sequence in the previous experiments as it showed the highest negative transfer among the 3 sequences, and measured the average success rate of each task after all tasks were learned. Figure 12 illustrates the results. Note that $|\mathcal{D}_\tau|$ and $|\mathcal{M}^k|$ indicate the size of replay and expert buffer respectively. Both SAC and PPO algorithms show that as the replay buffer size increases, the average success rate also increase. This is because if the size of the replay buffer is too small, the total number of samples used for training the model decreases, leading to insufficient learning. When we varied the size of the expert buffer, we did not observe any noticeable differences. Based on this result, we can reduce the expert buffer size to achieve better memory efficiency.

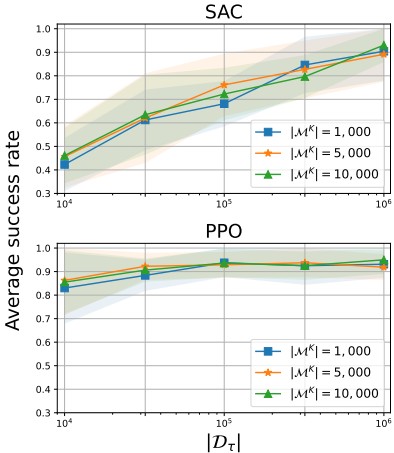

Figure 12: The average success rates of R&D with SAC and PPO on various $|\mathcal{D}_\tau|$ and $|\mathcal{M}|$.

# H  Efficiency of buffer generation in R&D

A potential concern with applying R&D is the reliance on expert rollout for distillation, which may increase training costs. While additional rollouts do contribute to the overall computational requirements, their impact is minimal compared to the training time of the online learner.

In our experiments, training SAC on a single task for 3 million steps required approximately 8 hours, whereas generating the buffer through expert rollouts took only about 15 minutes, accounting for roughly 3% of the total training time. This indicates that the buffer construction process is relatively efficient and does not significantly hinder overall training.

Furthermore, off-policy RL algorithms such as SAC inherently utilizes a replay buffer during online learner training, which can be directly repurposed for the distillation process in R&D. While this approach may lead to slight

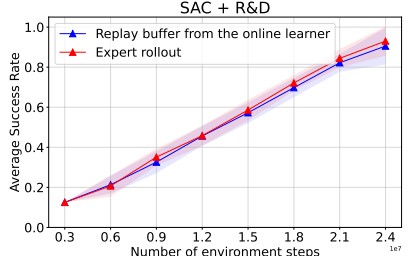

Figure 13: Comparison of performance on Hard sequence between using the replay buffer from the online learner (blue) and generating a new buffer through expert rollouts (red) in SAC + R&D.

performance degradation compared to constructing a separate buffer through expert rollouts, the difference remains marginal. As shown in Figure 13, the results indicate that reusing the replay buffer provides a viable alternative with comparable effectiveness, potentially reducing the need for additional expert rollouts.

# I EXPERIMENTS ON DEEPMIND CONTROL SUITE (TASSA ET AL., 2018)

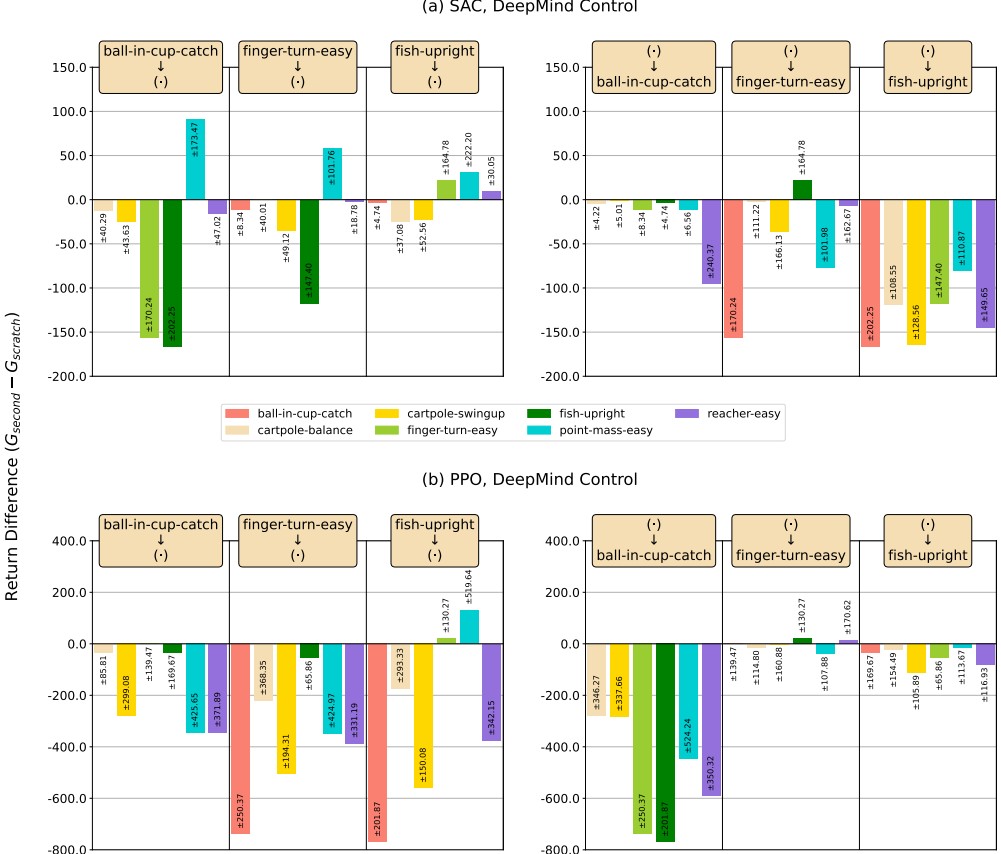

Figure 14: Two-task fine-tuning results for (a) SAC and (b) PPO with standard deviation. The values with a ± sign refer to the standard deviation.

To show the existence of the negative transfer in other domain, we also carry out experiments on DeepMind Control Suite Tassa et al. (2018). First, we select 7 tasks {ball-in-cup-catch, cartpole-balance, cartpole-swingup, finger-turn-easy, fish-upright, point-mass-easy, reacher-easy}. Those tasks are carefully selected which can be successfully learned from scratch within 1M steps. Different from the experiment in Section 3.1, we do not make groups on those tasks. After selecting the tasks, we also carry out the two-task CRL experiments on 18 pairs like in Figure 4 with 5 different random seeds. Figure 14 shows the results. In this case, similar to the case in Meta World experiment, the negative transfer in PPO is much severe than SAC. When the three tasks are in first tasks, the negative transfer occurs more frequently. Only ball-in-cup-catch is getting worse when it lies on the second task. In terms of SAC, there are some cases where the negative transfer occurs rarely or severely. For example, for the fish-upright task, the phenomenon is quite opposite when it lies on the first task (rarely occurs) or the second task (frequently occurs). For the other tasks, we can also find the negative transfer quite often. Therefore, also in the DeepMind Control Suite environment, we can easily find the negative transfer problem.

## I.1 RESULTS OF R&D ON DEEPMIND CONTROL SUITE (TASSA ET AL., 2018)

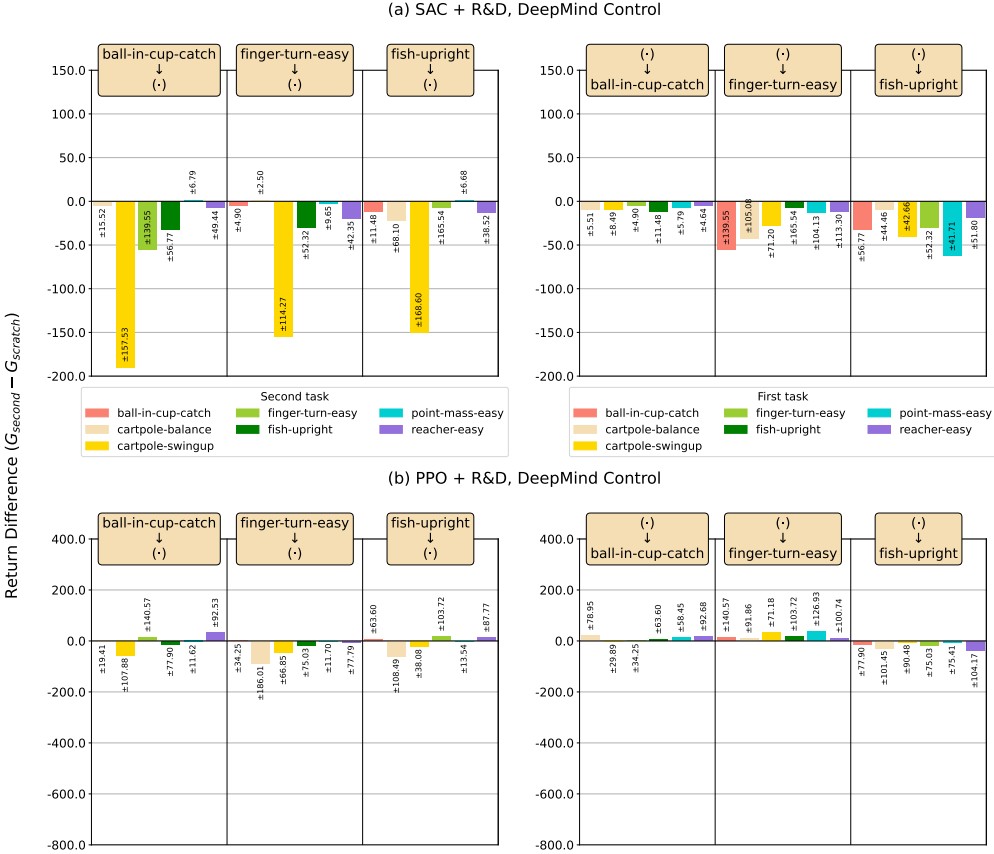

Figure 15: Two-task R&D results for (a) SAC and (b) PPO with standard deviation.

To further investigate the effectiveness of R&D for tackling the negative transfer in other domain, we carry experiments on DeepMind Control Suite Tassa et al. (2018). The overall experiment is same as in Section I. In Figure 15 (a), in most scenarios, our observations indicate that the R&D framework effectively alleviates negative transfer when compared to fine-tuning. However, an exception was noted when R&D was utilized in conjunction with SAC, leading to an unexpected performance decline in a specific task (cartpole-swingup). To investigate this phenomenon further, we conducted additional experiments. In these experiments, we employed knowledge distillation by transferring knowledge from a policy trained on the cartpole-swingup task to a randomly initialized agent. The results revealed a return difference of -209.1 ± 149.1, aligning closely with the performance observed when the offline actor was pre-trained on other tasks without resetting. These findings imply that the inability to learn the cartpole-swingup task is attributed to factors unrelated to negative transfer during the distillation process. If negative transfer were the underlying cause, applying knowledge distillation to a randomly initialized network would not have led to performance degradation relative to training the agent from scratch.

## J    EXPERIMENTS ON ATARI GAMES (MNIH ET AL., 2013)

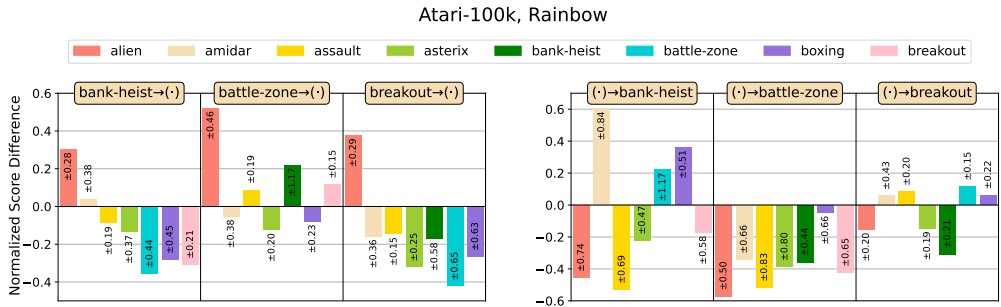

Figure 16: Two-task fine-tuning results for Rainbow with standard deviation. The values with a ± sign refer to the standard deviation.

To show the existence of the negative transfer in visual domain, we carry out experiments on Atari games (Mnih et al., 2013). First, we select 8 games {`alien`, `assault`, `bank-heist`, `boxing`, `amidar`, `asterix`, `battle-zone`, `breakout`}. In this experiment, we trained Rainbow (Hessel et al., 2018) on two task pairs, and we selected 3 representative tasks, {`bank-heist`, `battle-zone`, `breakout`}. Same as the experiment on DeepMind Control Suite, we did not make group in this experiment. We carry out two-task CRL experiment on 21 task pairs with 5 different random seeds. Figure 16 shows the result. First, in this figure, we can observe that the negative transfer frequently occurs across various task pairs. For example, when the tasks `bank-heist` and `breakout` lie on the first task, the most of the second tasks perform poorly. Furthermore, in case of `battle-zone`, the negative transfer pattern is opposite when `battle-zone` lies on the first task or the second task. For the former case, the degree of the negative transfer is small. However, for the latter case, most of the proceeding tasks suffer from the negative transfer severely.

## K    ANALYSIS ON THE ORDER OF THE KL DIVERGENCE IN R&D

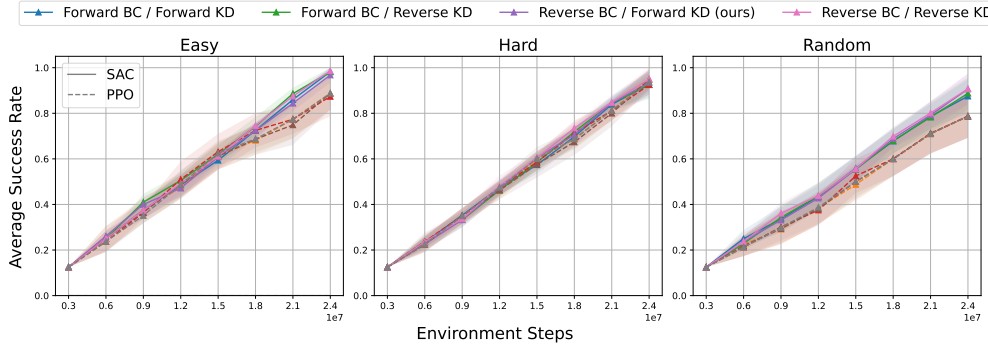

Figure 17: Results of experiments changing the direction of KL divergence for the R&D loss across three 8-task sequences (Easy, Hard, Random).

One may observe a difference in the orders of KL divergence between (a) and (b). This discrepancy arises from the fact that the order employed in (a) adheres to the traditional form of knowledge distillation Hinton et al. (2015), while the order in (b) follows the convention used in Wolczyk et al. (2022). In this section, we performed experiments by changing the direction of KL divergence for the R&D loss. These experiments were applied to the three 8-task sequences mentioned in the main text. Figure 17 shows the results. In the figure, the results indicate that the order of KL divergence did not significantly impact the performance.

# L  THE RESULTS WITH ERROR BARS

In this section, we report the results of Table 1, Figure 4, and Figure 7 with error bars, which corresponds to Table 2, Figure 19, and Figure 18, respectively.

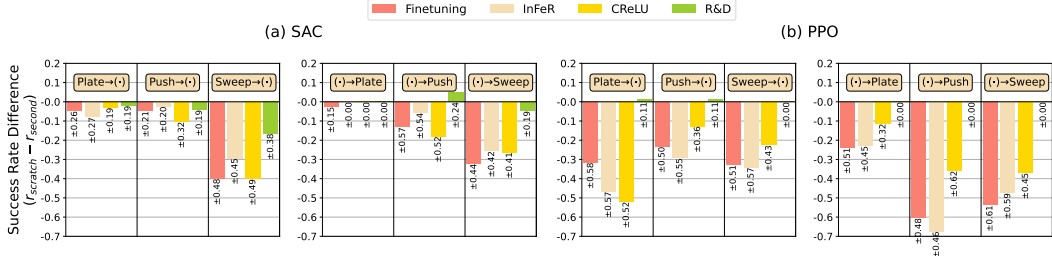

Figure 18: Two-task CRL experiments on various methods. Note that for the methods with CReLU, the results of 'From scratch' are obtained by training vanilla RL methods with CReLU.

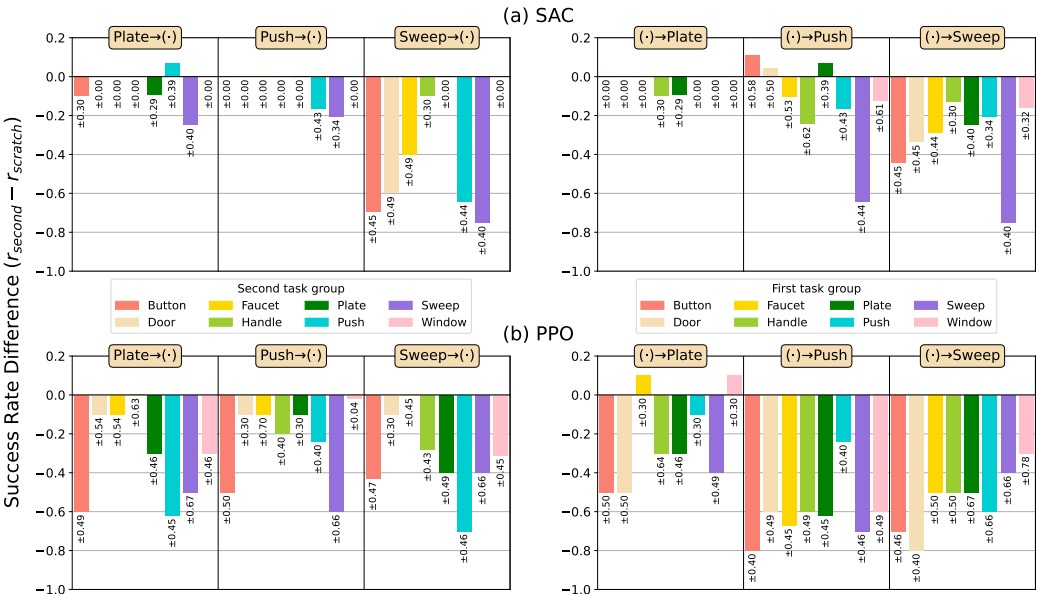

Figure 19: Negative transfer patterns for the two-task fine-tuning with (a) SAC and (b) PPO, when tasks from Plate, Push, Sweep groups are learned as the first (left) or the second (right) task. The values with a ± sign refer to the standard deviation.

Table 2: The transfer and forgetting results with standard deviation. Note that the numbers after ± represent the standard deviation.

| Measure | Transfer (↑) | | | Forgetting (↓) | | |
|---|---|---|---|---|---|---|
| Sequence | Easy | Hard | Random | Easy | Hard | Random |
| | SAC | | | | | |
| Fine-tuning | -0.0955 ± 0.0929 | -0.5002 ± 0.1236 | -0.1925 ± 0.132 | 0.8997 ± 0.0912 | 0.5040 ± 0.1333 | 0.7766 ± 0.1111 |
| EWC | -0.0708 ± 0.0813 | -0.4567 ± 0.0915 | -0.2598 ± 0.1294 | 0.8517 ± 0.1129 | 0.5123 ± 0.0969 | 0.6714 ± 0.1327 |
| P&C | -0.0708 ± 0.1134 | -0.5065 ± 0.1439 | -0.2077 ± 0.1517 | 0.8714 ± 0.1187 | 0.4723 ± 0.1338 | 0.7023 ± 0.1335 |
| ClonEx | -0.0570 ± 0.0768 | -0.5130 ± 0.1574 | -0.2760 ± 0.1322 | 0.0146 ± 0.0437 | 0.0049 ± 0.0632 | 0.0397 ± 0.0714 |
| ClonEx + CReLU | -0.1958 ± 0.1936 | -0.5580 ± 0.1166 | -0.2132 ± 0.1947 | 0.0389 ± 0.0557 | 0.0671 ± 0.0997 | 0.0117 ± 0.0291 |
| ClonEx+InFeR | -0.1172 ± 0.1030 | -0.5032 ± 0.1654 | -0.2322 ± 0.1655 | 0.0311 ± 0.0626 | 0.0006 ± 0.0666 | 0.0377 ± 0.1073 |
| R&D | -0.0020 ± 0.0232 | -0.0412 ± 0.0566 | -0.0140 ± 0.0603 | 0.0000 ± 0.0000 | 0.0083 ± 0.0359 | 0.0454 ± 0.0701 |
| | PPO | | | | | |
| Fine-tuning | -0.3788 ± 0.1866 | -0.6238 ± 0.1439 | -0.4250 ± 0.2318 | 0.3614 ± 0.1114 | 0.3314 ± 0.1117 | 0.3357 ± 0.1567 |
| EWC | -0.5363 ± 0.2493 | -0.6763 ± 0.1365 | -0.3750 ± 0.1250 | 0.3186 ± 0.1250 | 0.2814 ± 0.1591 | 0.4300 ± 0.0043 |
| P&C | - | - | - | - | - | - |
| ClonEx | -0.4250 ± 0.1785 | -0.6075 ± 0.1576 | -0.4375 ± 0.2183 | 0.0271 ± 0.0621 | 0.0429 ± 0.0655 | 0.0143 ± 0.0429 |
| ClonEx + CReLU | -0.325 ± 0.1392 | -0.6100 ± 0.1814 | -0.2750 ± 0.1458 | 0.0286 ± 0.0571 | 0.0029 ± 0.0086 | -0.0143 ± 0.0769 |
| ClonEx+InFeR | -0.0750 ± 0.1696 | -0.4625 ± 0.2440 | -0.2875 ± 0.3115 | 0.0429 ± 0.0655 | -0.0143 ± 0.0429 | 0.0000 ± 0.0000 |
| R&D | 0.0250 ± 0.0500 | 0.0250 ± 0.0500 | 0.0125 ± 0.0375 | 0.0500 ± 0.0906 | 0.0286 ± 0.0571 | 0.0286 ± 0.0571 |

# M    DETAILS ON EXPERIMENT SETTINGS

In the all experiments, we used Adam optimizer and the code implementations for all experiments are based on Garage proposed in Yu et al. (2020). For the machines, we used 16 A5000 GPUs for all experiments.

## M.1    HYPERPARAMETERS FOR THE EXPERIMENTAL RESULTS

The hyperparameters for SAC and PPO are described in Table 3 and Table 4, respectively. For the hyperparameters on the CRL methods, the details are described as follows:

- EWC, P&C: The regularization coefficient was set to 1000
- BC: The regularization coefficient was set to 1, and the expert buffer size $|\mathcal{M}_k|$ was set to $10k$ for task $k$.
- R&D: The regularization coefficient was set to 1, and the expert buffer size $|\mathcal{M}_k|$ was set to $10k$ for task $k$. Furthermore, the replay buffer size $|\mathcal{D}|$ was set to $10^6$

Table 3: Model hyperparameters for SAC

| Description | Value (Meta World) | Value (DeepMind Control) |
|---|---|---|
| **General Hyperparameters** | | |
| Maximum episode length | 500 | 1000 |
| Environment steps per task | 3M | 1M |
| Evaluation steps | 100k | 100k |
| Gradient updates per environment step | 1 | 0.25 |
| Discount factor | 0.99 | 0.99 |
| **Algorithm-Specific Hyperparameters** | | |
| Hidden sizes | $(256, 256)$ | $(1024, 1024)$ |
| Activation function | ReLU | ReLU |
| Policy learning rate | $3 \times 10^{-4}$ | $1 \times 10^{-4}$ |
| Q-function learning rate | $3 \times 10^{-4}$ | $1 \times 10^{-4}$ |
| Replay buffer size | $10^6$ | $10^6$ |
| Mini batch size | 64 | 1024 |
| Policy min. std | $e^{-20}$ | $e^{-20}$ |
| policy max. std | $e^2$ | $e^2$ |
| Soft target interpolation | $5 \times 10^{-3}$ | $5 \times 10^{-3}$ |
| Entropy coefficient($\alpha$) | `automatic_tuning` | 0.2 |

Table 4: Model hyperparameters for PPO

| Description | Value (Meta World) | Value (DeepMind Control) |
|---|---|---|
| **General Hyperparameters** | | |
| Maximum episode length | 500 | 1000 |
| Environment steps per task | 3M | 1M |
| Mini batch size | 64 | 1024 |
| Evaluation steps | 100k | 100k |
| Gradient updates per environment step | 1 | 1 |
| Discount factor | 0.99 | 0.99 |
| **Algorithm-Specific Hyperparameters** | | |
| Batch size | 15000 | 10000 |
| Hidden sizes | $(128, 128)$ | $(1024, 1024)$ |
| Policy activation function | ReLU | ReLU |
| Value activation function | tanh | tanh |
| Policy learning rate | $5 \times 10^{-4}$ | $3 \times 10^{-4}$ |
| Value learning rate | $5 \times 10^{-4}$ | $3 \times 10^{-4}$ |
| Policy min. std | 0.5 | 0.5 |
| Policy max. std | 1.5 | 1.5 |
| Likelihood ratio clip range | 0.2 | 0.2 |
| Advantage estimation | 0.95 | 0.95 |
| Entropy method | `no_entropy` | `no_entropy` |
| Normalize value input / output | True | True |

## N  SOCIETAL IMPACTS

The R&D method effectively addresses the negative transfer problem, significantly enhancing the performance and adaptability of AI systems. This improvement allows AI to learn new tasks more effectively without detrimental effects from previous experiences, leading to more robust applications. Industries reliant on AI for automation and optimization can benefit from increased efficiency and cost savings, as AI systems reduce downtime and the need for retraining. Additionally, advancements in robotics (e.g., healthcare robots, autonomous vehicles, and industrial robots) can lead to safer and more reliable robots, enhancing their integration into everyday and high-stakes environments.

## O  LIMITATIONS

Though, in our work, we only consider the effect of the negative transfer, considering the positive transfer is also important point in CRL. Our method, R&D, can effectively resolve the negative transfer, but does not have the ability on the positive transfer by utilizing the useful information on the previous tasks. Furthermore, our experiments are mainly focused on Meta World environment, and we did not carry out experiments on much larger scale such as Atari or Deepmind Lab.

