# OpenReview forum: "Prevalence of Negative Transfer in Continual Reinforcement Learning: Analyses and a Simple Baseline"
_ICLR.cc/2025/Conference — ICLR 2025 Poster_

### Official Review · Reviewer_xWNm · 2024-10-25

**Soundness:** 3
**Presentation:** 4
**Contribution:** 2
**Rating:** 6
**Confidence:** 4

**Summary:**

The paper proposes Reset and Distill (R&D), a method inspired by policy distillation [1] and neural network growth [2], to address catastrophic forgetting in knowledge transfer. R&D operates by reusing expert samples from previous tasks to transfer knowledge to subsequent ones while resetting the online learner network’s knowledge. The authors demonstrate, through comprehensive experiments, that the sequence of tasks [3] significantly affects performance when learning multiple tasks.

*[1] Hinton, Geoffrey. "Distilling the Knowledge in a Neural Network." arXiv preprint arXiv:1503.02531 (2015).*

*[2] Rusu, Andrei A., et al. "Progressive neural networks." arXiv preprint arXiv:1606.04671 (2016).*

*[3] S. Narvekar, J. Sinapov, and P. Stone, “Autonomous task sequencing for customized curriculum design in reinforcement learning,” in (IJCAI), The 2017 International Joint Conference on Artificial Intelligence, 2017.*

**Strengths:**

* *Comprehensive literature review*: The authors provide a thorough and well-structured review of recent research related to plasticity loss in non-stationary environments and negative transfer in transfer learning. This positions the paper well in the context of existing research.

* *Robust experiments*: The empirical validation is solid, with experiments conducted on recognized benchmarks in the RL community. These experiments effectively test the assumption that the arrival order of tasks impacts learning performance, and R&D outperforms previous methods in mitigating catastrophic forgetting, providing valuable insights.

**Weaknesses:**

* *Similarity with P&C*: In Figure 3, the concept of an "adaptor" within the Progress & Compress (P&C) framework is not well explained. While the term "reset" may be somewhat intuitive, it is not clearly detailed in either the main text or the Appendix. Given that the novelty of R&D is acknowledged to be marginal compared to P&C, a more in-depth discussion of how these concepts differ is necessary to strengthen the paper's contribution.

* *Sequence of tasks focus*: The paper emphasizes that the arrival order of tasks have impact in negative transfer, yet the proposed method does not directly address this issue. Instead, R&D relies on known techniques (distillation and resetting) to mitigate catastrophic forgetting.


* *Contradictory hypothesis*: The authors propose that *all prior knowledge from previous tasks should be erased* when learning a new task. However, this approach seems contradictory, as retaining certain previous knowledge can, in some cases, enhance learning, especially when tasks share high similarity.

**Questions:**

- [Q1] If task sequencing is feasible, could curriculum learning enhance R&D performance by optimizing task order?

- [Q2] It’s unclear why mitigating negative transfer does not necessarily imply facilitating positive transfer, and vice versa.

- [Q3] Could you provide more details about why Proximal Policy Optimization (PPO) appears more susceptible to negative transfer than Soft Actor-Critic (SAC)? Given the comparison between these algorithms, understanding the underlying reasons could offer valuable guidance for developing more robust approaches to continual learning.

---

> ### Author Response · Authors · 2024-11-21
> **Response to the comments (Part I)**
>
> ### Weakness 1: A more in-depth discussion of how P&C and R&D are different is necessary as the novelty of R&D is marginal compared to P&C
>
> #### 1. Regarding insufficiently explained concepts in P&C
>
> The term "reset" in the context refers to re-initializing the parameters of a neural network. In P&C, two modules exist: the active column, which learns the current task, and the knowledge base, which retains information from previous tasks. In this context, reset specifically re-initializes only the parameters of the active column. This concept is not new, as the original P&C paper also discusses the possibility of resetting the active column.
>
> In P&C, the adaptor is an additional mechanism designed to enable the active column to utilize past information stored in the knowledge base. Specifically, the adaptor combines the activations of the active column and the knowledge base in a layer-wise manner, as expressed by the following equation:
>
> $$
> h_i = \sigma \left( W_i h_{i-1} + \alpha_i \odot U_i \sigma \left( V_i h_{i-1}^{\text{KB}} + c_i \right) + b_i \right)
> $$
>
> Here, $W_i$ and $b_i$ represent the weight and bias of the active column, while $U_i$, $V_i$, and $c_i$ denote the weight and bias of the adaptor. To compute the activation $h_i$ of the active column, the activation $h_{i-1}^\text{KB}$ from the knowledge base is processed through the adaptor.
>
> Although these points are explained in the original P&C paper, they are particularly relevant to the experiments illustrated in our Figure 3. Therefore, we agreed that a more detailed explanation was necessary. We plan to organize this appropriately and include it in the appendix.
>
> #### 2. Regarding the key differences between R&D and P&C
>
> It is true that the online/offline actors mechanism in R&D is similar to P&C's implementation of the active column/knowledge base. However, there are not only architectural differences between P&C and R&D but also fundamentally significant differences in their underlying motivations.
>
> P&C is a continual learning algorithm designed with inspiration from Progressive Neural Networks (PNN) [1]. A key feature of both algorithms is the utilization of models trained on previous tasks through lateral connections. To achieve this, adaptor modules, as described above, are extensively employed to facilitate these connections. This is not merely a structural detail; the adaptors serve as essential components for leveraging knowledge acquired from previous tasks to effectively learn the current task, thereby enabling *progress*. In this regard, the knowledge base in P&C is not only a continual learner but also a supporting mechanism that enhances the efficient learning of the active column.
>
> In contrast, R&D does not utilize adaptors or other lateral connections. This indicates that there is no transfer from the offline actor to the online actor, as such transfer can sometimes negatively impact the learning of the current task. We have demonstrated this experimentally in Figure 3. As a result, unlike P&C's knowledge base, the offline actor in R&D functions solely as a continual learner, receiving knowledge distillation from the online actor.
>
> While there are differences between R&D and P&C, as described earlier, we do not wish to strongly assert the originality of R&D. In addition to its similarities with P&C, R&D has a structural limitation in that it cannot facilitate positive transfer. In this regard, R&D serves merely as a simple baseline capable of addressing negative transfer in a simple manner. Our contribution lies in demonstrating through experiments that negative transfer occurs frequently in CRL settings, and that addressing this issue with a simple method like R&D can significantly enhance the performance of the agent and even outperform state-of-the-art baselines.
>
> Again, this is an important aspect of understanding R&D, and we plan to more thoroughly include above arguments in our final manuscript.
>
> [1] Rusu et. al., Progressive Neural Networks, arXiv preprint, 2016
>
> ### Weakness 2: The order of tasks has impact in negative transfer, yet R&D does not address this issue directly
>
> As you mentioned, we believe that the order of tasks significantly influences the negative transfer. If it were possible to actually optimize the order, it could lead to faster learning in R&D. ***However, altering the order of the task sequence in continual learning appears to be inappropriate, as the problem setting assumes that access to tasks to be learned in the future is not available.*** Thus, we believe that combining R&D with sequence scheduling is somewhat infeasible and out of scope.
>
> A detailed explanation of this can be found in our response to [Q1]. Please refer to it for further details.

---

> ### Author Response · Authors · 2024-11-21
> **Response to the comments (Part II)**
>
> ### Weakness 3: Contradictory hypothesis
>
> We do not deny the possibility of positive transfer occurring in continual reinforcement learning. The reason R&D resets the online learner for each task is not because all prior knowledge is detrimental, but rather because it is challenging to identify the specific circumstances under which negative transfer occurs. Admittedly, this approach may not be the most optimal. However, we have attempted to demonstrate experimentally that this naive approach can effectively eliminate the potential for catastrophic negative transfer.
>
> ### Question 1: Could curriculum learning enhance R&D performance by optimizing task order?
>
> While it is an intriguing idea, there are significant challenges to implementing curriculum learning effectively in the context of R&D. To construct an efficient task sequence through curriculum learning, access to detailed information about all tasks is typically required. However, in the setting of continual learning, this assumption is unnatural. Continual learning focuses on how agents can learn sequentially without having access to information about tasks that will be encountered in the future or tasks that have already been learned.
>
> Furthermore, even if direct access to future tasks is not required to design a curriculum, adjusting the task sequence through a curriculum in the context of continual learning appears to be somewhat impractical. This is because, as previously mentioned, continual learning inherently assumes that access to tasks other than the one currently being learned—whether previously learned tasks or tasks to be learned in the future—is not available, making it impossible to modify the sequence order. For this reason, applying curriculum learning in continual reinforcement learning (RL) may prove to be practically infeasible.
>
> ### Question 2: It’s unclear why mitigating negative transfer does not necessarily imply facilitating positive transfer, and vice versa.
>
> Until now, negative transfer has not been considered as a significant topic in the field of continual RL. Many continual RL methods have instead evolved in a direction where knowledge acquired from previous tasks is utilized in the current task[2, 3]. This can be interpreted as assuming that positive transfer occurs and seeking to leverage it. In such a situation, if negative transfer were to occur, the knowledge acquired from previous tasks could potentially interfere the training on current task. In this case, existing methods designed to facilitate positive transfer may fail to address negative transfer and could even exacerbate the issue.
>
> [2] Mendez et. al., Modular Lifelong Reinforcement Learning via Neural Composition, ICLR, 2022
>
> [3] Wolczyk et. al., Disentangling Transfer in Continual Reinforcement Learning, NeurIPS, 2022
>
> ### Question 3: Why PPO appears more susceptible to negative transfer than SAC?
>
> Thank you for pointing this out.
> While we do not have rigorous analyses on this issue, we propose the following hypothesis. PPO is an algorithm that improves upon TRPO[4] and is based on the trust region method. This implies the presence of a constraint that prevents the policy from deviating excessively from the policy of the previous step during learning. We hypothesize that due to this constraint, or the surrogate loss derived from it, the policy may be restricted from significantly deviating from previously learned outcomes, potentially exacerbating negative transfer. In contrast, SAC appears to exhibit a relatively reduced effect of negative transfer, as its algorithm is not influenced by the policy of the previous step.
>
> [4] Schulman et. al., Trust Region Policy Optimization, arXiv preprint, 2015

---

> > ### Comment · Reviewer_xWNm · 2024-11-26
> >
> > I appreciate the authors’ efforts to thoroughly address both the strengths and weaknesses of their proposed method during the rebuttal phase. The paper provides a clear illustration of how task arrival impacts knowledge transfer, particularly within the unique constraints of the Continual Learning setting, where task sequencing cannot mitigate these challenges. Furthermore, the results and insights presented in the study are compelling. Taking these factors into account, I have decided to update my score.

---

> ### Author Response · Authors · 2024-11-25
> **A gentle reminder for feedback**
>
> Dear Reviewer xWNm,
>
> Since the discussion period is almost approaching to its end, we would greatly appreciate your additional feedback on our response to your review. Would you please let us know whether our response has helped you further clarify our paper?
>
> Thank you very much in advance.

---

### Official Review · Reviewer_MLHD · 2024-10-28

**Soundness:** 4
**Presentation:** 4
**Contribution:** 3
**Rating:** 8
**Confidence:** 4

**Summary:**

Author addresses the challenges of negative transfer in Continual Reinforcement Learning (CRL). Negative transfer occurs when learning new tasks negatively impacts performance, which is problematic for CRL systems that aim to handle multiple tasks sequentially without forgetting. The authors identify that existing CRL methods often focus on mitigating plasticity loss or promoting positive transfer, but these methods fail to effectively counteract negative transfer.
They propose a new baseline method called Reset & Distill (R&D) to address negative transfering problem. R&D works by resetting the agent’s online networks for each new task while preserving learned knowledge through distillation into an offline model. and this dual-actor setup, which allows an online network for current learning and an offline network for cumulative knowledge,enableing R&D to handle sequential tasks more effectively by preventing both negative transfer and forgetting. Experiments in their work demonstrate R&D outperforms other models on task success rates, emphasizing the need to prioritize countering negative transfer in CRL​, which seems promising

**Strengths:**

Firstly, it focuses on an important yet overlooked issue,the occurrence of negative transfer in Continual Reinforcement Learning (CRL). This problem has a significant impact in practical applications, but prior research has largely concentrated on preventing forgetting or promoting positive transfer, often neglecting negative transfer. Authorr introduces the "Reset & Distill" (R&D) method specifically to tackle this issue, demonstrating a strong sense of innovation and hitting the problem "right on target." The experimental design is also solid, verifying the prevalence of the negative transfer issue across different environments, such as Meta World, DeepMind Control Suite, and Atari, which adds credibility to the results. Additionally, the paper is written in a clear manner, particularly in explaining the design and implementation of the R&D method, making it easy for readers to understand the authors' approach. Overall, this paper brings fresh perspectives and practical solutions to the CRL field

**Weaknesses:**

While the paper does a good job of demonstrating the effectiveness of the R&D method through extensive experiments, it somewhat lacks a deep theoretical analysis explaining why this approach is effective at mitigating negative transfer. What are the fundamental mechanisms that allow R&D to sidestep the pitfalls of negative transfer?

**Questions:**

① Balancing Negative and Positive Transfer: The R&D method seems to heavily focus on preventing negative transfer by resetting the online learner's parameters. While this is a novel approach, I wonder about the potential for R&D to leverage positive transfer. Is there a way to modify R&D to harness the knowledge from previous tasks without incurring negative transfer? Your thoughts on this balance would be insightful.
② Theoretical Underpinnings: The paper does a great job showcasing the empirical success of R&D, but I must say, I'm left wanting more in terms of theoretical analysis. What are the theoretical reasons behind R&D's effectiveness in reducing negative transfer? How does the periodic reset and distillation process interact with the learning dynamics at a deeper level? A bit more depth here could really help solidify the contribution of your work.
③ Computational Cost: The dual-network architecture of R&D, with periodic resets and distillation, seems promising. However, I'm concerned about the computational overhead this might introduce, especially for resource-constrained settings. Could you discuss the computational efficiency of R&D and any potential strategies to mitigate these costs?

---

> ### Author Response · Authors · 2024-11-21
> **Response to the comments**
>
> ### Question 1: Is there a way to modify R&D to harness the knowledge from previous tasks without incurring negative transfer?
>
> Thank you for the important question. Indeed, if R&D can be improved to also facilitate positive transfer, the algorithm would become more general and robust. An intuitive approach one can come up with is to reset the online actor _only when_ the negative transfer is anticipated. If such prediction method of negative transfer can be realized, we can prevent the negative transfer only when necessary and achieve the benefit of the positive transfer in other cases.
>
> However, accurately predicting the extent of transfer between tasks at high-level can be challenging. For example, sequentially learning `door-close` after `door-open` enjoys a positive transfer; i.e., the agent can quickly reach the door handle since the two tasks are inherently similar. But, such transfer does not happen when learning `sweep-into`  after `sweep`, despite the two tasks also seem similar, and a severe negative transfer occurs.
>
> We may therefore attempt to devise a proxy measure for the transferability at the early stage of training on each task and use that to determine whether to reset or reuse the networks. In Section 3, we showed that the indicators of the plasticity loss cannot be such faithful proxies, hence we need to find some alternatives, e.g., gradients of cumulative rewards at the early learning stage, etc. Rigorously defining such proxy and evaluating to improve R&D would be what we can pursue as future research.
>
> ### Question 2 & Weakness 1: What are the theoretical reasons behind R&D’s effectiveness in reducing negative transfer?
>
> Thank you very much for the constructive comment. While we definitely believe that a more thorough theoretical analysis on the effectiveness of R&D would be very valuable, it seems to be quite a challenging task at this point -- rigorously defining similarity between different RL tasks is extremely hard. While leaving the rigorous theoretical analysis as future work, we re-iterate our intuitive reasoning that is given in our manuscript (line 419~421, line 455-457). Namely, we hypothesize that changing the training scheme from bootstrapping to supervised learning may give an advantage on tackling the negative transfer. Let assume that the previously learned agent causes the negative transfer on the following task. If we still stick to training the agent using bootstrapping, the biased policy and critic cannot escape from the region causing the negative transfer. However, by changing the training scheme to imitation learning (as in R&D), the supervision from the expert can promote learning the following tasks. In our future research work, we will try to take theoretical analysis on the above explanation.
>
> ### Question 3: Computational efficiency of R&D and any potential strategies to mitigate these costs
>
> In our experiment, we use RTX A5000 GPU for all experiments. In this setting, training online SAC agent for 3M environment steps in Meta World takes 8 hours. For training the offline agent, extracting the episode rollouts using the online agent takes 15 minutes, and training the agent in offline manner during 50 epochs takes 25 minutes. Therefore, the additional time for training the offline agent is only 40 minutes, and we think it is somewhat marginal compared to the online training.
>
> The straightforward way to reduce the computation cost is as follows. First, reducing the number of epochs for offline training can improve the efficiency. In our expeirment, we check that using only 1~10 epochs for offline training can resolve the negative transfer, and the performance drop is quite marginal compared to the case using 50 epochs. The second way is reusing the replay buffer collected from the training phase of online actor. Though reusing the replay buffer can reduce the performance, the degree of the degradation is not significant.

---

> > ### Comment · Reviewer_MLHD · 2024-11-24
> >
> > Thank you for the detailed and thoughtful responses to my questions. The clarifications provided have resolved my concerns, and I now have a much clearer understanding of the contributions and methodology. I believe the paper is ready for acceptance.

---

### Official Review · Reviewer_PBSr · 2024-10-31

**Soundness:** 3
**Presentation:** 3
**Contribution:** 3
**Rating:** 8
**Confidence:** 3

**Summary:**

Authors demonstrate the existence of negative transfer in CRL setup and show that this issue can not be mitigated with previous CRL approaches. Authors also propose a solution to the problem which demonstrates strong reduction of the negative transfer effect

**Strengths:**

1. Authors reveal the existence of the negative transfer effect which might strongly affect the further research in CRL field. Further studies might take this aspect into consideration when designing novel approaches or applying existing for tasks of their interest.
2. The presence of the negative transfer is demonstrated across 3 different domains  (Meta World, DM and Atari) which provide significant evidence that the negative transfer might be a frequent effect.
3. Using a wide range of techniques it is shown that promoting positive transfer cannot address negative transfer.
4. The possible solution (R&D) for mitigating the negative transfer is presented. It it compared against multiple CRL approaches and shown to be effective.
5. Ablation study on hyperparamters effect for R&D is conducted.

**Weaknesses:**

1. While R&D reduces negative transfer effect, sometimes it suffers from forgetting more than other approaches.
2. When applied with SAC, R&D requires a vast amount of expert rollouts to keep the good performace.
3. R&D performance is shown only on Meta World tasks if I understand correctly.

**Questions:**

I don't have any major concerns about this work. The only interesting thing for me would be to see the R&D performance on DM Control and Atari domains.

---

> ### Author Response · Authors · 2024-11-21
> **Response to the comments**
>
> ### Weakness 1: R&D sometimes suffers from forgetting more than other approaches
>
> Thank you for your comment. We believe this comment is based on the results in Table 1. In the table, it is true that the average of the forgetting measure for R&D is sometimes higher than those of other baselines (primarily ClonEx and its variants) in certain cases. However, when considering the variances of those results, we believe the level of forgetting of R&D is roughly at the same level as ClonEx and its variants: values around 0.05. Note this is an order-of-magnitude lower level than those of other CRL baselines, i.e., Fine-tuning/EWC/P&C. Thus, we believe it is reasonable to assert that both R&D and ClonEx effectively address the forgetting issue in CRL. This is not surprising since R&D and ClonEx employ the same method (behavioral cloning) to prevent forgetting.
>
> The more detailed results, including their statistical significances, are in in Table 2 of Appendix A.8. If you have any additional questions, please let us know.
>
> ### Weakness 2: R&D applied with SAC requires a vast amount of expert rollouts to keep the good performace
>
> It is true that additional rollouts in SAC can increase the overall training cost. However, the time required to generate a buffer through rollouts for offline learner training is not significantly large compared to the time spent training the online learner. In our experiments, training SAC on a single task for 3 million steps took approximately 8 hours, whereas constructing the buffer through rollouts required only about 15 minutes, corresponding to roughly 3% of the total training time for the online learner. This demonstrates the efficiency of the buffer generation process relative to the overall training duration.
>
> Moreover, since SAC employs a replay buffer during the training of the online learner, it is possible to directly use this buffer in the distillation process of R&D. While this approach may result in a slight performance degradation compared to generating a new buffer through rollouts, the difference is not substantial. In our experiment, the average success rate for the 'Hard' sequence was 0.93 (±0.07) when a new buffer was constructed using expert rollouts, compared to 0.88 (±0.12) when the replay buffer from the online learner was directly reused. This suggests that reusing the replay buffer provides a viable alternative with comparable effectiveness.
>
> We apologize for lacking full details on the computational cost in our original manuscript. We sincerely appreciate your constructive feedback and will incorporate above information in the final manuscript.
>
> ### Weakness 3 & Question 1: R&D performance on DM Control and Atari domains
>
> Thank you for your valuable feedback. We agree that verifying the robustness of R&D requires a thorough explanation of whether R&D operates effectively across different environments. To address this concern, we conducted experiments, as illustrated in Figure 5, by applying R&D to SAC and PPO within the DeepMind Control environment. The detailed results of these experiments can be found in Figure 14 of Appendix A.6.1.
>
> In most cases, we observed that R&D effectively mitigates negative transfer compared to finetuning. However, when R&D is applied with SAC, we identified an unexpected performance drop in a specific task (`cartpole-swingup`). To analyze this in more depth, we conducted additional experiments. Specifically, we applied knowledge distillation from a policy trained on the `cartpole-swingup` task to a randomly initialized agent. The results obtained from this experiment yielded a return difference of -209.1±149.1, which is comparable to the values reported when the offline actor was pre-trained on other tasks without being reset. These findings suggest that the failure to learn the `cartpole-swingup` task arises from factors other than negative transfer during the distillation phase. If negative transfer were the cause, then applying knowledge distillation to a randomly initialized network should not have resulted in any performance degradation compared to training from scratch.
>
> For the Atari experiments, we used the Rainbow algorithm for training. Rainbow employs a distributional RL method [1], which poses difficulties in defining distillation between different Q-functions. To address this challenge, the R&D framework could potentially employ alternative imitation learning techniques instead of distillation. If an imitation learning method applicable to distributional RL settings becomes available, it could be utilized. However, implementing such a solution is unlikely to be feasible within the timeframe of this discussion period. We express our apologies for this limitation. Once again, we thank you for your constructive comments and hope that these findings contribute to your understanding.
>
> [1] Bellemare et. al., A Distributional Perspective on Reinforcement Learning, ICML, 2017

---

> > ### Comment · Reviewer_PBSr · 2024-11-25
> >
> > I thank authors for their response. I will increase my score

---

### Official Review · Reviewer_U6a3 · 2024-11-04

**Soundness:** 2
**Presentation:** 2
**Contribution:** 2
**Rating:** 6
**Confidence:** 2

**Summary:**

This work considers negative transfer problem in continual RL, where steam of RL tasks may severely hurt the policy's performance. The authors proposed to a method named Reset & Distill (R&D) for mitigating such issue. R&D stores  a copy of single-task policy as offline policy for each task in the task stream and also a subset of data of that task; and it then optimize the KL divergence between online policy and all offline policies on the stored data of prior tasks, and the KL divergence between online policy and offline policy of current task.

**Strengths:**

- Method is simple and plausible

- Empirical evaluation appears strong

**Weaknesses:**

- I find the paper is a bit difficult to follow as a lot of concepts are described quite verbally such as negative/positive transfer and also the baselines -- although I am not familiar with the literature of continual RL, there might be room for clarity improvement for general RL audience with continual learning background

- Despite I like the visualization of Figure 2, and I understand the R&D is introduced after that, it might be more visually clear how R&D perform if the authors could also include R&D in Figure 2.

---

While I think continual RL could be an interesting area and the method proposed could be plausible to mitigate the negative transfer issue, the overall writing is somewhat verbal, hence slightly difficult to follow especially for people working in general RL. Besides, I am not sure if this evaluation is a standard evaluation protocol in the CRL community. Therefore, I find this paper is boarderline to me although with low confidence.

**Questions:**

- As I am not particularly familiar with the literature, I wonder do the baselines store copies of policies and data. If some of them have the same setting, storing policies and data, what are the corresponding budget, compared to R&D.

- Is the sequential evaluation of Meta World a standard benchmark in the CRL literature or it is customized in this work? (If it is customized, is there a commonly used CRL evaluation benchmark in the literature?)


---

post-rebuttal: 5 -> 6.

---

> ### Author Response · Authors · 2024-11-21
> **Response to the comments (Part I)**
>
> ### Weakness 1: Lots of concepts and baselines are described verbally
>
> Thank you for your comment. While there have been some attempts on mathematically defining the notions of positive/negative transfers [1, 2, 3], exactly quantifying those notions has been challenging and their definitions were different from each other. Conceptually, however, the amount of transfer can be measured by comparing the performance of a transfer-learned model to that of a model learned scratch, which is reflected in our metric $T_\tau$ given in Section 5.3. We hope this quantitative metric, which we mainly used in Section 5, is clear and intuitive to the reviewer. Furthermore, we will polish our description on the notions in our final version so that it could be made as clear as possible.
>
> Regarding the baselines, we had to briefly describe them due to the space constraint -- we will include more in-depth descriptions of them in the Appendix. In a nutshell, EWC and P\&C are widely used continual learning baselines for both supervised and reinforcement learning. ClonEx is designed to promote the positive transfer in continual reinforcement learning by using task-specific exploration and behavioral cloning. Moreover, as we discussed in Section 2.2, InFeR and CReLU are the methods for tackling the negative transfer by regularizing the features of penultimate layer and replacing the ReLU activation with CReLU activation, respectively.
>
> We hope above descriptions on the baselines, which will be further elaborated in the Appendix, are helpful for the clarification.
>
> [1] Wang et. al., Characterizing and avoiding negative transfer, CVPR, 2019
>
> [2] Cao et. al., Partial transfer learning with selective adversarial networks, CVPR, 2018
>
> [3] Ge et. al., On handling negative transfer and imbalanced distributions in multiple source transfer learning, Statistical Analysis and Data Mining: The ASA Data Science Journal, 2014
>
> ### Weakness 2: Result of R&D in Figure 2
>
> Thank you for your constructive comment. Although we believe including the performance of R&D in Figure 2 would not necessarily align with the flow of the paper, since we introduce R\&D later and the purpose of Figure 2 is to highlight the inability of previous methods on addressing the negative transfer problem, we agree that it would be interesting and informative to show the performance of R&D on the exact same exemplary tasks. In the Appendix A.4, we inclued not only the results of R&D but also the result of the online actor which corresponds to 'w/ reset w/o adaptor'. In Figure 11, we can observe that the success rate of R&D in `push-wall` is close to the results of the online actor, and the online actor does not suffer from the negative transfer compared to other variations. Therefore, through the above result, we want to stress that resetting the whole agent and discarding the previous knowledge can mitigate the negative transfer problem effectively.
>
>
> ### Weakness 3 & Question 2: Not sure if this evaluation is a standard evaluation protocol in the CRL community
>
> Thank you for the clarification question. We note that several mainstream previous works on CRL has also adopted the sequential evaluation protocol as ours on the well-established benchmark environments such as Meta World, DM Control and Atari. For example, [3] introduced a new benchmark protocol for CRL, dubbed as Continual World, by sampling multiple tasks from Meta World and evaluating the sequential learning capability on them. Their follow-up work [4] adopted this benchmark as well. Moreover, [5,6] also adopted the same evaluation protocol on the locomotion tasks that are highly similar to those in DM Control to evaluate the performance of their proposed CRL algorithms. Furthermore, [1] and [2] also took the same evaluation protocol on the Atari environment. Therefore, we argue that the sequential evaluation protocol is quite standard.
>
> [1] Schwarz et. al., Progress & Compress: A scalable framework for continual learning, ICML, 2018
>
> [2] Rolnick et. al., Experience Replay for Continual Learning, NeurIPS, 2019
>
> [3] Wołczyk et. al., Continual World: A Robotic Benchmark For Continual Reinforcement Learning, NeurIPS, 2021
>
> [4] Wołczyk et. al., Disentangling Transfer in Continual Reinforcement Learning, NeurIPS, 2022
>
> [5] Kaplanis et. al., Continual Reinforcement Learning with Complex Synapses, ICML, 2018
>
> [6] Kaplanis et. al., Policy Consolidation for Continual Reinforcement Learning, ICML, 2019

---

> > ### Author Response · Authors · 2024-11-21
> > **Response to the comments (Part II)**
> >
> > ### Question 1: Budgets of baselines compared to R&D
> >
> > We stress that the difference of the memory budgets between the baselines and R&D is marginal. Namely, both EWC [1] and P&C [2] also store the networks that learned previous and current tasks, while ClonEx [3] stores state samples for computing the behavioral cloning loss. Usually, storing the samples takes much larger memory budget than storing the network parameters. For R&D, it stores both two models and state samples in the buffer. In case of the memory budget, compared to ClonEx, the additional component in R&D is the network for offline actor, and in our experiment, the number of parameters for the networks we used is small. Therefore, R&D does not require large amount of memory budget compared to the ClonEx. In case of the computational budget, training the online actor for all methods takes 8 hours. For R&D, extracting the rollouts takes 15 minutes and the training time for offline actor takes 25 minutes. Therefore, the additional computational buget for R&D is also small compared to the baselines.
> >
> > [1] Kirkpatrick et. al., Overcoming catastrophic forgetting in neural networks, PNAS, 2017
> >
> > [2] Schwarz et. al., Progress & Compress: A scalable framework for continual learning, ICML, 2018
> >
> > [3] Wołczyk et. al., Disentangling Transfer in Continual Reinforcement Learning, NeurIPS, 2022

---

> ### Comment · Reviewer_U6a3 · 2024-11-25
>
> Thank you for the detailed responses. My concerns on evaluation protocol and budgets have been partially addressed hence I am updating my evaluation to 6.

---

### Meta-Review · Area_Chair_G4dA · 2024-12-20

**Metareview:**

This paper studies the problem of negative transfer in continual reinforcement learning (CRL). The authors first demonstrate the existence of this problem and propose Reset&Distill (R&D) which consists of resetting the network followed by distilling the policy from the prior task. The method is evaluated on both PPO and SAC and compared against relevant baselines.

The paper is well-written and the problem well-formulated and motivated. The proposed method is simple yet clear, and appears to be effective at reducing negative transfer.

The method seems to be derived mostly from intuition and the success of R&D setup seems to be sensitive to the order of tasks. There is also a shared concern on the limited novelty of the method, as it relies on pre-existing ideas.

However, given the thorough empirical evaluation and relatively unexplored problem setting, I recommend accepting this work, as it may prove useful for future work to build on.

**Additional Comments On Reviewer Discussion:**

There were concerns raised on clarity of presentation and thoroughness of empirical evaluations, which the authors mostly addressed during the rebuttal. Given the improvements and discussion with authors, there is general agreement on acceptance.

---

### Decision · Program_Chairs · 2025-01-22

Accept (Poster)